# Child public health interventions for conflict-affected populations: A systematic review

Ayesha Kadir[1], Sneha Krishnan[2,3], Catherine McGowan[4*], Daniel Martinez Garcia[5,6]

1 Department of Paediatrics, Viborg Regional Hospital, Denmark, 2 Jindal School of Public Health and Human Development, OP Jindal Global University, Sonipat, Haryana, India, 3 Environment Technology and Community Health Consultancy Services, Mumbai, India, 4 Department of Infectious Disease Epidemiology and International Health, London School of Hygiene and Tropical Medicine, London, United Kingdom, 5 Institute of Global Health, Faculty of Medicine, University of Geneva, Geneva, Switzerland, 6 TACTIC Project, Medical Department, Médecins sans Frontières, Geneva, Switzerland

* Catherine.McGowan@lshtm.ac.uk

## Abstract

Armed conflict causes pervasive harm to children, and humanitarian responses to support them face significant challenges. This review aims to summarise the evidence on the effectiveness of interventions to treat, protect, and promote child public health in conflict-affected populations. A systematic review was performed, with searches of major databases and the grey literature from 1 January 2012 – 20 February 2025. Included studies provided data on child or caregiver outcomes associated with interventions to support children affected by armed conflict. Studies on nutrition and perinatal interventions were excluded. Data were extracted on the setting, population, intervention design, study type, and findings. The searches yielded 3,601 records. 51 intervention studies met inclusion criteria, 39% of which were trials. Studies were mainly from Africa (51%), the Middle East (25%) and Asia (18%). The majority of studies focused on mental health and psychosocial support (MHPSS) (N = 29, 57%). MHPSS, child protection, and/or parenting interventions were the focus of trials as well as intersectoral interventions. Somatic child health interventions (N = 19, 37%) focused on immunisation, adolescent sexual and reproductive health, toxic stress, and telemedicine services. Five studies measured development outcomes and one intervention targeted children with disabilities. Over half of the studies were carried out amongst displaced populations. Intervention design varied widely within and between sectors. Studies showed promising results, particularly for non-specialist MHPSS interventions. Only 20% of studies assessed intervention safety. The evidence for child public health interventions to support conflict-affected populations is increasing, with increased numbers of studies over time, and improved study design, execution, and reporting. However, the evidence remains poor, limited to a few topic areas and with continued geographical disparities. There are notable gaps in evidence on the safety of interventions, their medium- and long-term impacts,

**Data availability statement:** All data are provided within the manuscript or supplementary information files.

**Funding:** The authors received no specific funding for this work.

**Competing interests:** The authors have declared that no competing interests exist.

sustainability, and interventions for child development and children living with disabilities.

## Introduction

Armed conflict is known to cause pervasive harm to children, including physical, psychological, and social injuries that can impact their health, psychomotor development, and life trajectories. The scale of the impact of armed conflict on children is large: 473 million children (or more than 1 in 6) live within 50 km of an active conflict zone, and 76% live in a country that is affected by armed conflict [1,2].

Conflict can impact children directly through combat activities, and indirectly through a range of pathways, including but not limited to destruction of infrastructure, forced displacement, lack ability to meet basic needs, and changes in social arrangements, all of which place children at heightened risk for experiencing violence, deprivation, and exploitation [3]. The literature on conflict and health describes an extensive range of harms to children involving every organ system [3–6]. In addition to illness, injury, disability and death, children may be orphaned, and they may take on adult responsibilities such as working and/or caring for family members [3]. The destruction of infrastructure, including direct attacks on health care, create barriers in access to care and place patients and families at risk of injury, illness, and death while seeking care [3,4]. Disruption of communities, breakdown in social arrangements, and displacement of populations are associated with increased risk of exploitation and abuse of children [3,4]. Child labour, physical abuse and sexual abuse have been documented in conflicts across the globe [2]. Conflicts also affect children who live far from where the fighting occurs. For example, family members may participate in or be directly affected by armed conflict; children living in other countries may witness news, social media imagery, and discourse about conflicts, attacks and violence; and they may experience economic and social changes that are related to increasingly internationalised conflicts [3].

According to international law, children in conflict settings hold special rights to protection and support [7–9]. Yet, the year 2023 saw the highest number of verified grave violations against children since records began [2]. The six grave violations against children in armed conflict are: the killing and maiming of children, the recruitment and use of children, sexual violence against children, the abduction of children, attacks on schools and hospitals, and denial of humanitarian access [10]. These violations have been recorded repeatedly in conflicts across the world [2]. At the time of writing, the Palestinian population is being subjected to genocide by Israel [11,12]. Nearly 2/3 of documented deaths in Gaza from traumatic injuries are in women, children and the elderly [13], and humanitarian relief has been restricted to the point of famine [14]. The United Nations has reported medicide by the Israel Defense Forces, through the targeted destruction of the Gazan health system with an intent to destroy medical care [15]. In Sudan, children have been targeted in killings, abductions, and sexual violence, and more than 5 million children have been forcibly displaced since the civil war began in April 2023 [16]. Attacks on schools, hospitals, and entire

villages, the hindrance of humanitarian aid, and recruitment of children to armed groups have been documented [16]. These are just two examples from among more than 120 active armed conflicts globally [17].

Humanitarian response is organised into sectors that address different needs of the population, including health, nutrition, water, sanitation, and hygiene, protection, shelter, education, food security and livelihoods, social and behaviour change, and cash and voucher assistance. In order to improve the safety and effectiveness of humanitarian response, a range of standards, tools, and guidance have been developed, such as the SPHERE Standards and sector-specific tools and guidance [18–23]. Efforts have also been made to identify aspects of the Sustainable Development agenda that address the needs and context in humanitarian settings [24,25]. While some of the available tools address the specific risks and needs of children in different ages and stages of development, the evidence base remains limited.

Recent reviews describing the effects of armed conflict on child health suggest that conflict affects children through nuanced and context-specific ways [3,6,26]. However the nuances remain poorly understood, thus limiting our understanding of the kinds of interventions that are appropriate, safe, and effective to mitigate harm and improve child health. There is marked geographic disparity in which populations of children are studied [3], and routine data on child public health in conflict settings are limited [27]. As such, we have sought to improve our understanding of the effectiveness and safety of recent interventions to support child health in conflict settings.

We report below the findings from a systematic review of intervention studies aimed at mitigating harm and improving child health. We report against items in the updated PRISMA reporting guidelines (S1 Text) [28]. The review protocol is registered with PROSPERO (CRD42022356007).

## Methods

The exposure of interest was direct or indirect experience of armed conflict, including displacement due to conflict, living in an area where active combat is occurring or has occurred within the previous generation, or having a first-degree relative who has been directly affected by armed conflict. We relied on author determinations of indirect exposure. Studies presenting data on child public health outcomes relating to conflict or interventions to support children impacted by conflict were included in the original review.

Database searches were conducted in MEDLINE, EMBASE and Global Health (all via Ovid) on 30 August 2022 using variations of terms for children, armed conflict, and humanitarian or crisis. The protocol, including the full search terms are available from PROSPERO (CRD42022356007). Searches were limited to sources published since 1 January 2012. No language restrictions were applied. Duplicates were removed in EndNote [29] and the remaining records were uploaded to Covidence [30] for title/abstract screening and full text review.

Grey literature searches were conducted in DuckDuckGo, ReliefWeb, and in the websites of Médecins Sans Frontières, Save the Children International, International Rescue Committee and UNICEF on 27-28 September 2022 using the search terms "child" and "armed conflict". The first 100 records retrieved from each search were screened on title/abstract, and subsequently on full text. Additional records were identified from review of citation lists during full text screening. Screening and full text review were completed independently by two reviewers. Discrepancies in screening outcomes were resolved through discussion. A third author resolved any persisting discrepancies.

We carefully considered carrying out a risk of bias assessment but ultimately opted not to. Conflict settings place extreme limits on public health research. Security risks, barriers in access to affected populations, and movements of populations typically compromise study rigour and introduce bias. However, omitting or otherwise reducing the importance of studies in conflict settings on the basis of study quality would, paradoxically, introduce bias towards less severe conflict settings and compromise external validity. Moreover, subjecting studies to a risk of bias assessment would almost certainly have required us to redefine the scope of our review. Finally, our review presents a typology of interventions, and intervention designs, aimed at improving the health and wellbeing of conflict-affected children; the identified studies did not enable meta-analysis or other quantitative synthesis. The application of gold standard criteria for studies in

conflict-affected populations limits the availability of evidence and undermines the value of evidence from the hardest to reach populations [27,31]. Lack of data renders these populations invisible, limits the ability of humanitarian actors to deliver safe and effective interventions, and hampers progress and improvement in research among conflict-affected populations. Taken together, the justification for carrying out a risk of bias assessment or a certainty assessment was weak when compared to the benefits of not doing so.

Studies were included if they were undertaken in humanitarian settings caused by armed conflict and reported substantively on interventions targeting children <18 years of age or their caregivers. Studies were included if they reported on any of the following outcomes: morbidity, disability, child development, child mental health, caregiver mental health, changes in social behaviours, schooling, and/or mortality. Papers that provided non-disaggregated data that included children <18 years as well as young people up to <24 years were included. In addition to original research articles, agency technical reports were included if a clear description of the study methods was provided. No restrictions were made for language or geographic location. Military studies were excluded, as these studies report on a special population that sought care from a military health facility, and are thus unlikely to represent the general population of children affected by conflict [3]. Studies that only provided data on nutrition, perinatal, or neonatal interventions and outcomes were excluded, due to recent reviews on these subjects [5,32,33].

An extraction form was created in Microsoft Excel [34] and included information about: setting, intervention design, type of conflict, location of study, study design, outcomes, and limitations. Two authors supported data extraction, one of whom reviewed the entire dataset for internal consistency in data extraction. A template data extraction form (S1 Table) and analytic codes (S2 Text) are available in the supplementary materials.

One author updated the MEDLINE and EMBASE searches on 20 February 2025, screened the titles using the original criteria but limited to intervention studies only, and extracted the findings for studies meeting inclusion criteria. A second author reviewed the dataset of the search update for internal consistency, including review of the full text papers.

The extracted data on intervention studies from the original review and the search update were compiled in Microsoft Excel, coded and analysed for trends in study design, geographic location, thematic focus, target group age and gender, displacement status, and whether the study mentioned exploration of any harm caused to children from the intervention. As the included studies varied markedly in focus, design, and reporting of outcomes, we were unable to carry out a meta-analysis.

## Results

The database searches retrieved 3094 records (Fig 1). After removing duplicates, 2108 records were screened, of which 235 were sought for retrieval. One full text paper was not available. 234 papers were reviewed in full text, of which 109 were excluded. The grey literature searches retrieved 501 records. Six additional papers were identified by hand searches of citations. After removing duplicates, 460 records were assessed for eligibility from the grey literature and citation searches. Our final sample included 51 intervention studies. The included studies and their main findings are presented in Table 1.

The general characteristics of included studies are presented in Table 2. Twenty interventions were studied in trials, seventeen of which were randomised controlled trials. The trials were almost exclusively focused on mental health and psychosocial support (MHPSS) (N = 16/51) and/or child protection (N = 9/51) and seven included a parenting component. Fourteen trials were conducted with displaced populations. Other study designs included mixed methods (N = 12/51), before and after observational (N = 8/51), cross-sectional (N = 4/51), qualitative (N = 3/51), descriptive (N = 2/51), cohort (N = 1/51) and case-control (N = 1/51). All three qualitative studies were focused on child protection, parenting, and/or MHPSS interventions. All of the studies that included measures of access to services focused on somatic paediatric health interventions.

Approximately half of studies were undertaken in Africa (N = 26/51) [35–60], followed by the Middle East (N = 13/51) [61–73], Asia (N = 9/51) [74–82], and Europe (N = 1/51) [83]. Two studies were of multi-centre interventions, one with

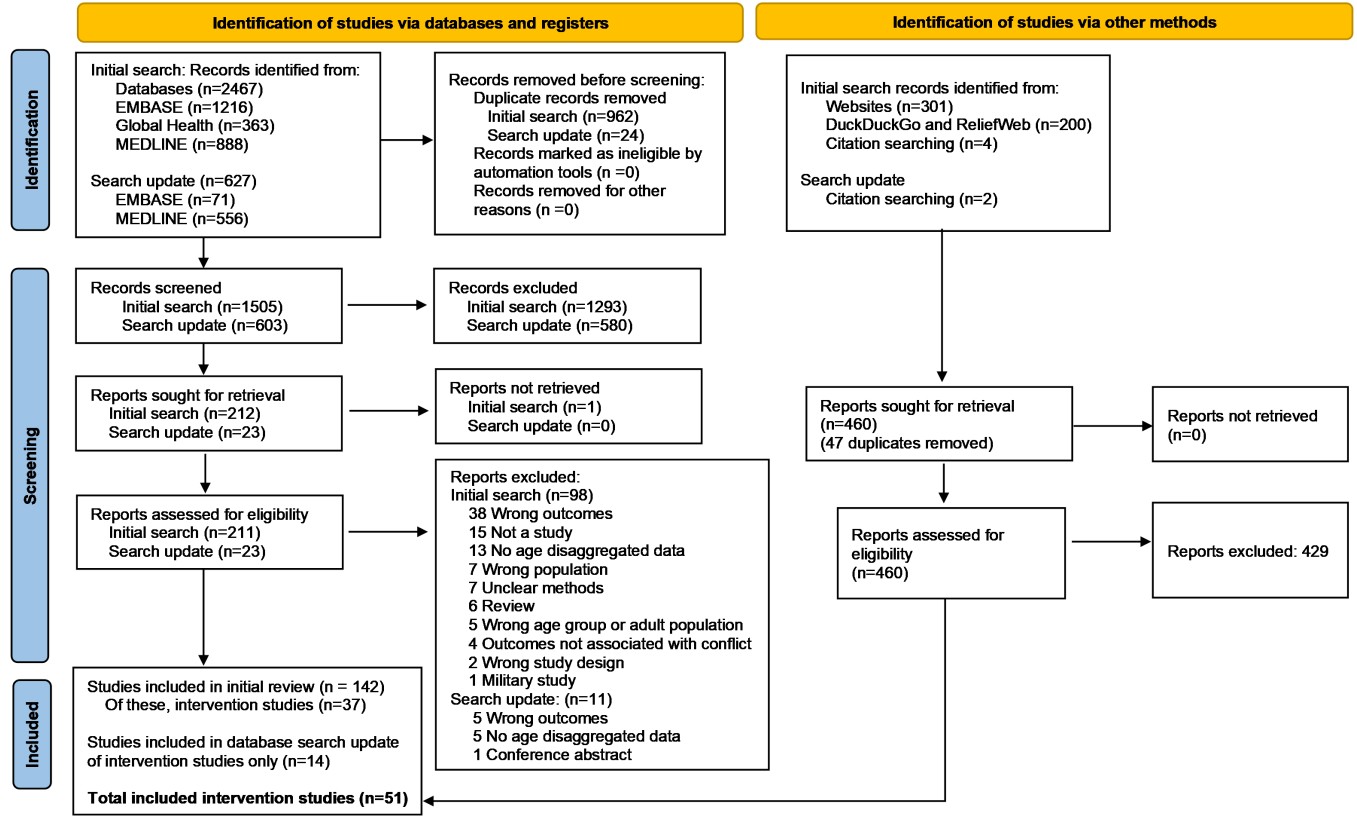

**Fig 1. PRISMA flow diagram.**

sites in Africa, the Middle East and Asia [84], and the other with a global focus (the Monitoring and Reporting Mechanism (MRM) on Grave Violations Against Children in situations of Armed Conflict) [85]. Fig 2 presents the geographical distribution of included studies. No intervention studies were published in the Americas, despite armed conflicts occurring in ten countries in these regions during the period 2012–2024 [86].

Studies tended to focus on short-term intervention effects beginning immediately post-intervention, and up 16 months after the intervention was stopped (N = 50/51). The longest follow up period was for a microfinance intervention aimed at improving adolescent MHPSS, development, and protection outcomes in the Democratic Republic of the Congo (DRC) – it was followed up for two years [40].

Over half of the studies focused on displaced populations (N = 28/51): 11 studies amongst internally displaced populations [35,42–44,47,52–54,58,59,77,82] and 17 amongst refugees [38,39,49,50,56,57,61,63,66–71,78,83,84]. Studies of internally displaced people most often included the host population (N = 8/11) [35,43,44,47,52,53,59,82], whereas refugee studies tended to focus on refugees only (N = 12/17) [38,39,49,50,56,57,61,63,71,78,83,84]. Nearly half (N = 8/17) of the refugee studies described interventions in Lebanon and Jordan [61,63,66–71].

## Mental health and psychosocial support interventions

Over half of the included studies focused on MHPSS (N = 29/51) [37,38,40,47,49,50,56,57,59,61,63,64,66–77,79–81,83,84]. Child protection (N = 20/51) [39,40,42,46,49,50,55–58,66,67,69,70,72,77,79,83–85] and parenting support (N = 15/51) [38,42,54,55,57,58,61,63,64,67,68,72,79,80,83] were overlapping areas of focus with studies frequently measuring

**Table 1. Included studies.**

| Publication | Geographical region of the intervention | Population | Study design | Sample size | Intervention | Summary of findings relating to children |
|---|---|---|---|---|---|---|
| Abdullahi et al (2020) [35] | Nigeria | IDPs and host communities in 12 local government areas and 12 control local government areas | Case-control | 13,316 children and 270,239 adults | Active case finding intervention for TB combined with HIV testing in IDP communities and control communities. Suspected cases were linked to treatment. Treatment was integrated into existing health clinics established by other organizations in the IDP camps, with some additional support to facilities as part of the intervention. | 283,556 verbal screening for TB encounters among the IDP population, including 270,239 TB adults and 13,316 children. A total of 1,423 people were diagnosed with TB and 1,419 were started on treatment. TB findings are not age disaggregated. 821 children eligible for TB prophylaxis (TPT): 552 (67.3%) received TPT and 409 (74%) completed the course. Rifampicin resistance identified in 2 paediatric TB patients. 13% of HIV tests performed were in children <15 years. HIV prevalence in IDP children <15 years was 0.9%, compared to the national rate of 0.2% |
| Loko Roka et al (2014) [46] | Democratic Republic of the Congo | Survivors of sexual violence, all ages | Retrospective cohort study | 671 | Sexual violence services in a remote active conflict zone in (Masisi) and a remote post-conflict setting (Niangara). A comprehensive package of care included a full medical examination including a genital and/or anal examination, optout HIV counselling and testing and pregnancy testing, emergency contraception for all females aged 12–45 who presented within 120 hours after rape, prophylaxis for sexually transmitted infections, HIV post-exposure prophylaxis for all rape survivors presenting within 72 hours, hepatitis B and tetanus vaccination, wound care, psychological counselling, preparation of a medico-legal certificate, medico-legal support if requested, and safe shelter and external referral for social assistance for specific cases in Masisi. Services were free of charge. Messaging on sexual violence care and health promotion was done in both communities. | In Niangara, the median age in years at the time of assault was 15 years (IQR 13–17). Only 32% of patients arrived within 72 hours and 42% arrived within 5 days of the assault. The main reason for delay in presentation was fear (22%). Referrals were mainly from NGOs (30%), community activities (11%) and theatre/drama (17%), respectively. 95% cases were rape, 2% sexual touching, 1% forced to rape, and 3% nonsexual aggression. The perpetrator was most often known to the survivor (48%). Physical trauma was more frequent in adolescents (N=69, 38%). In Masisi, patients were mainly adult survivors, 60% arrived within 72 hours and 63% arrived within five days. The main reason for delayed presentation was lack of knowledge on the available treatment. Most survivors were self-referred, referred through a friend, or from community activities. Both sites had high coverage of most components of sexual violence care given at first visit (PEP initiation, STI prophylaxis, emergency contraception, pregnancy testing). There was low coverage of follow up activities (HIV testing, completion of PEP, completion of vaccination). |
| Liddle et al (2013) [45] | Somalia | Patients receiving TB treatment | Cross-sectional study | 6167 patients, not disaggregated to provide number of children | The intervention included TB screening and treatment in outpatient clinics and therapeutic feeding centres or specific TB clinics. Treatment was given according to WHO-recommended standardised short course chemotherapy. Adherence was promoted by DOT providers carrying out TB health education activities in the DOT centre, the TB village and the inpatient ward. A monthly nutrition package was provided for patients to support their health and to promote adherence and contract was signed with patients to complete treatment, with a guarantor who was given a dry food ration as an incentive to support the patient until treatment completion. DOT corners and TB village was implemented in places where access was problematic. | The majority of paediatric cases had smear-negative pulmonary TB. Diagnosis of extrapulmonary TB increased with age until 15 years and then dropped again. Children <1 year old were less likely to have successful outcomes compared with older children and adults: 63.7% [242/380] for children <1 year old; 79.7% [420/527] for 1 to <5 years old; 84.4% [341/404] for 5 to <15 years old; 80.0% [3174/3967] for ≥15 years; p<0.001). Being <1 year old at the start of treatment increased the odds of death compared with adults (adjusted OR 2.47, p=0.001). Admission of smear-positive overall patients significantly decreased over time; this was thought to be due to an increased proportion of children presenting for treatment in later years of the intervention. |

*(Continued)*

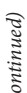

**Table 1.** (Continued)

| Publication | Geographical region of the intervention | Population | Study design | Sample size | Intervention | Summary of findings relating to children |
|---|---|---|---|---|---|---|
| Ngoy et al (2013) [51] | Somalia | Children receiving hospital care | Cross-sectional study | 6211 | Hospital services were provided free of charge. Services included paediatrics, internal medicine, maternity and emergency surgery. 36 hospital beds were allocated to children and newborns. Care for all services, including adults, was provided by four medical doctors and 80 nurses. Quality of care was evaluated by monitoring tool and routine outcomes data. | The study found that paediatric care provision is possible in remote conflict settings with severely limited access and that routine child health outcomes data can be collected and studied in these settings. Adverse outcomes (death and absconded) were within the acceptable threshold set by the organisation. 6166 children <15 years old were admitted between 2010–2011; 82% were <5 years old. The main reasons for admission were lower respiratory tract infections (48%) and severe acute malnutrition (16%), and these were also the leading causes of death. The highest case-fatality rate was in children with meningitis (20%). Violence accounted for 0.4% of admissions. The overall mortality rate was 3.3%. 2.8% of patients left the facility prior to discharge. 93.7% were cured at discharge, and 0.2% were transferred to another facility. |
| Njoh et al (2022) [52] | Cameroon | People eligible for vaccines in the Expanded Programme on Immunisation | Cross-sectional study | Not reported | Periodic intensification of routine immunization (PIRI) and disease surveillance were carried out in three rounds in each health district from March 2020 to January 2021. Routine immunisation was done door-to-door in communities and areas where IDPs lived, and in health facilities. The PIRI included three days of sensitization followed immediately by three days of vaccination and active case finding in the community for acute flaccid paralysis, and suspected cases of measles, yellow fever, and neonatal tetanus. | The intervention improved EPI coverage. 54,242 individuals received at least one catchup dose of a missed vaccine and the number of children vaccinated peaked in the months corresponding to the intervention. Every health district experienced an improvement in vaccination coverage compared to 2019. Penta-3 coverage increased from 43% in 2019 to 71.1% in 2020. Similar trends were observed for OPV-3 and IPV. MR-1 coverage increased from 43.2% in 2019 to 70.5% in 2020, and BCG coverage improved from 48.4% in 2019 to 73.2% in 2020. Disease surveillance improved for yellow fever, measles and AFP in 2020 compared with 2019. None of the implementing health workers was reported to have been hurt due to armed conflict or became ill with symptoms of COVID-19 as a result of their participation in the intervention. |
| Oladeji et al (2019) [53] | South Sudan | IDP and host community children <5 years | Cross-sectional study | 4,358 | A combined nutrition and vaccination intervention was implemented in communities and health facilities. Immunization services were integrated with nutrition services in outpatient therapeutic feeding (OTP) centres and nutrition outreach sites where immunization was previously not provided. Children <5 years seen at the OTP centres and during nutrition community outreach were assessed for their immunization status and vaccinated as appropriate. Mass nutrition screening by Community Nutrition Volunteers (CNVs) included assessment of immunization status before provision of nutrition services, and appropriate vaccination was provided. CNVs were trained and supported to include health education and immunization promotion messages, nutrition education, and counselling. | The combined immunisation-nutrition programme increased vaccination coverage of BCG, OPV3, Pentavalent 3 and measles. A high proportion of vaccinations were done in the OTPs and significantly lower dropout rates were observed among children receiving immunisations through OTP. Children vaccinated at the OTP centre in one site were 45% less likely to miss vaccination than those vaccinated at the primary health centre (OR: 0.45; 95%CI: 0.36- 0.55, p<0.05), while those vaccinated at the other OTP site were 27% less likely to miss vaccination than those vaccinated at the primary health centre (OR: 0.27; 95%CI: 0.20 -0.35, p<0.05). |

*(Continued)*

| Publication | Geographical region of the intervention | Population | Study design | Sample size | Intervention | Summary of findings relating to children |
|---|---|---|---|---|---|---|
| Ghbeis et al (2018) [65] | Syria | Syrian children with intensive care needs | Descriptive | 19 | Paediatric intensive care telemedicine support by volunteer, civilian medical professionals for health care workers providing health care to children in two hospitals in Syria. Support was provided through Facebook Messenger and Skype. | Paediatric intensive care telemedicine support was successfully provided to support the care of 19 children at two hospitals. Patients ranged from 1 day - 11 years old. The focus of support included mechanical ventilation settings, fluid and medications, management of seizures and remote real-time direction of active resuscitation. |
| Martinez Torre et al (2022) [90] | Nigeria | IDP and host community patients receiving mental health services | Descriptive | 1365 patients (11.7%) <15 years old | Psychological and pharmacological care, counselling, focused psychosocial support groups, psychosocial stimulation, psychoeducation, recreational activities, and psychological first aid. MHPSS activities were delivered by community mental health workers and lay counsellors trained and supervised by clinical psychologists. Patients with severe mental health disorders were managed by a medical doctor trained in the World Health Organization mhGAP intervention approach as well as by a clinical psychologist, with remote supervision by a psychiatrist. | 1365 patients were ≤15 years (mean 11.6 years, range 1–15 years. Anxiety was the most common symptom (n=469, 34.5%), followed by depression (n=333, 24.5%), posttraumatic stress (n=271, 19.9%), somatoform (n=107, 7.9%), other (n=98, 7.2%), psychotic (n=43, 3.2%), behavioural (n=27, 2%), and cognitive symptoms (n=11, 0.8%). Only 45.2% of children improved with treatment. Over half (53.5%; n=283) showed no change at the conclusion care, and 7 (1.3%) children reported a worsening condition. 73% of adult patients reported improvement. Children had different outcomes based on the tool used to assess symptoms. The CGI-I scale showed improvement in somatoform symptoms (OR: 2.3, p<0.001); posttraumatic symptoms (OR: 2, p<0.001); anxiety (OR: 1.6, p=0.001); and depression (OR: 1.5, p=0.002). The MHGS scale in children only showed improvement in depression (OR: 1.4, p=0.015). |
| Asghar et al (2018) [77] | Pakistan | Adolescents aged 12–19 and their caregivers | Mixed methods | 78 girls | Creating Opportunities through Mentorship, Parental Involvement, and Safe Spaces (COM-PASS) program for adolescent girls aged 10–14 years and a caregiver of their choosing. The intervention included provision of safe spaces, life skills and social assets building, mentorship of girls, and engaging caregivers as support systems and advocates for girls. The intervention included a 12-month girls' life skills and empowerment program and separate caregiver curriculum. | The intervention was associated with significant increases in the number of places participants visited in the previous month (median 2, p=0.002). It also led to improved psychosocial wellbeing and social support, with significantly increased self-esteem (p<0.001) and hope scores (p=0.001), increased friendships outside of the family (OR 1.27, p=0.007), presence of a trusted non-familial female adult in their lives (OR1.63, p=0.041), and of knowing where to access assistance for physical violence (OR 1.62, p=0.002). Participants were more likely to agree with statements about girls working outside the home after marriage (p<0.001) and girls deserving the same opportunities as boys (p<0.001),but there were no changes in beliefs about access to education, age of marriage, or attitudes towards violence against girls and women. There were no significant changes in the number of places where girls felt safe, and girls did not report feeling more comfortable discussing program content with caregivers at post-test. Perceptions of safety were related to gender dynamics in public spaces and fears of confronting and experiencing violence from boys and men in public spaces. |

*(Continued)*

 

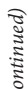

| Publication | Geographical region of the intervention | Population | Study design | Sample size | Intervention | Summary of findings relating to children |
|---|---|---|---|---|---|---|
| Dine et al (2023) [36] | NW and SW Cameroon | Young people aged 10–24 years and health care providers | Mixed methods | 114 | Sexual and reproductive health (SRH) services in the study regions | Sexual and reproductive health services were available in the study area, but young people (10–24 years old) were not always aware of the available services or how and when to access them. There was no specialized adolescent or youth SHR service. The available services are not designed for or with young people, and health workers identified a need for training and support to provide SRH services for this population. Outreach did not regularly include communication methods used by young people, such as social media, to provide SRHS information. Young people and health workers described additional barriers for young people in access to care, including shame, stigma, fear, lack of sexual and reproductive health information, and insecurity. |
| Dozio (2020) [38] | Cameroon | Pregnant or breastfeeding refugee women and their infants | Mixed methods | 1022 | A Baby Friendly Spaces (BFS) intervention comprised of seven 6m2 shelters in three refugee camps. BFSs were set up near women's shelters, with a space for individual consultation and an area for group activities. Support included clinical evaluation of psychological status on admission to the programme, breastfeeding counselling, complementary feeding counselling, baby massage, mother-child play sessions, relaxation exercises, psychosocial support, and support for group discussions on childcare practices, home visits and community awareness activities. A referral system to the Baby Friendly Spaces programme was established in collaboration with other agencies. | Women and their infants participated in the BFS for up to six months. Participants reported significant improvement in wellbeing and reduced psychosocial suffering (p = 0.000) and an increase in perceived social support (p = 0.000). Observed mother-infant interactions and relationships improved (p = 0.000). Women experienced a reduction in breastfeeding difficulties (p = 0.000). Length of time engaging with the BFS correlated positively with improved outcomes for reduced psychosocial suffering (p < 0.01), mother–infant relationships (p < 0.01), social support (p < 0.05), and reduced breastfeeding difficulties (p < 0.05). Women shared positive reflections on the BFS, describing improved social support and reassurance that they are not alone in their concerns and worries. |
| El-Khani et al (2021) [83] | Serbia | Primary caregivers of children 8–15 years old | Mixed methods | 25 | Strong Families" intervention, a family skills training programme comprised of 1–2 hour sessions for 3 weeks. Parallel sessions for children and their primary caregiver were followed by joint family sessions. Caregiver sessions explored challenges faced by caregivers, stress management, showing affection whilst setting boundaries, listening to children, encouraging good behaviour and discouraging misbehaviour. Child-specific sessions focused on coping with stress, exploring rules and responsibilities, reflecting on their goals, and reflecting on the roles their caregivers play in their lives. Family sessions focused on positive communication, stress relief techniques, family values and sharing appreciation. Facilitators came from local civil society organizations working in reception centres and were trained for the intervention. | At 6-week follow up, there was a significant reduction in Strengths and Difficulties Questionnaire measures for total difficulties (p = 0.002), emotional problems (p < 0.001), conduct (p = 0.001), and hyperactivity (p = 0.012). There were no intervention effects on peer problems and prosocial behaviour scores. Parenting and family adjustment measures improved for caregivers with higher baseline scores, reaching statistically significant improvement in coercive parenting, parent–child relationship and parental adjustment subscales. Families with lower parenting and adjustment scores did have significant changes any measures after 6 weeks. Parents described improved understanding of their children's behaviour and how to respond to it, reduced use of corporal punishment, improved self-control and emotional regulation, improved communication with their children and also their partners, and increased focus on the wellbeing of their children. |

*(Continued)*

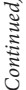

**Table 1.** (Continued)

| Publication | Geographical region of the intervention | Population | Study design | Sample size | Intervention | Summary of findings relating to children |
|---|---|---|---|---|---|---|
| Evans et al (2024) [79] | Afghanistan | Children with disabilities and their female caregivers | Mixed-methods realist-informed observational study | 118 | The "Mighty Children" programme, a participatory educational support group for caregivers of children living with disability. The programme was adapted from the Ubuntu programme to be more generally applicable for children with physical disabilities, delivered by non-health experts, and placing emphasis on caregiver psychosocial well-being. This intervention was facilitated by midwives and nurses. Nine sessions focused on caring practices for a child living with disability, nurturing care, psychological resilience skills, child rights, parenting strategies and peer support. Participants were female caregivers of children with disability. The children were also invited to attend the sessions and participate in the programme. | Caregivers reported improvement in their quality of life and family functioning after the programme, which was significant across all domains of the Paediatric Quality of Life, family impact module (PedsQL–FIM) questionnaire. Pashtun mothers had greater improvement than Hazara or other Persian caregivers (p = 0.012). The second cohort reported significantly larger improvements in quality of life than the first (p < 0.001); qualitative exploration suggested this was due to improved skills of facilitators and caregiver knowledge and familiarity with the programme. Caregivers described changes in mindset from frustration to value and love. Parenting practices shifted to increased patience, kindness and nurturing. Caregivers and families increased inclusive behaviours, including how they called the child ("mighty" vs "disabled") and observed increased participation of the child in family and social life. The programme improved psychosocial well-being for caregivers and also their children, reduced social isolation and a feeling of not being alone. Some caregivers reported their children showed improved mobility, communication, and self-care after they had participated in the programme. |
| Glass et al (2020) [40] | Democratic Republic of the Congo | Adolescents 10–15 years old | Longitudinal mixed methods | 542 | An integrated adult parent and young adolescent animal microfinance/asset transfer program that was set up at the request of parents engaged in the adult programme. The parent programme, Pigs for Peace (PFP), included village-based group training on raising and breeding healthy pigs, commitment to livestock care per project standards, and repayment on the pig asset loan. Participants were provided a vaccinated and healthy pig, monthly village-based group support meetings and home visits, and veterinary support as needed. When the participating household reimbursed the project with two female piglets, these repayment piglets were provided as an asset loan to participants on the waitlist. The adolescent programme, Rabbits for Resilience (RFR), was adapted from the adult programme with similar design but using rabbits. | The study compared caregiver-adolescent dyads (RFR + PFP), adolescents participating alone (RFR) and caregivers participating alone (PFP). At 24 month follow up, all three groups improved asset building scores (p = 0.002), school attendance (p = 0.015), and prosocial behaviour scores (p = 0.032). Adolescents in the RFR + PFP group and RFR group had greater improvement in prosocial behaviour, with the greatest effect seen in adolescents in RFR + PFP. School attendance was greater in the RFR + PFP group and the RFR group compared with caregiver participation alone. There were no differences over time in internalizing behaviours, reported experience of stigma, or food security. Differences in outcomes varied by age and gender for asset building, prosocial behaviour, school attendance, experienced stigma, and food security. Adolescents reported that the RFR program improved their ability to pay for school fees, help their families meet basic needs including food, and increased their status in their family and community. Challenges included death of rabbits, theft, and potential conflict within the household on how the rabbit asset should be used. |
| Kahow et al (2024) [41] | Somalia | Children <5 and pregnant women | Mixed methods | N/A | Mobile health clinic intervention, the Far-Reaching Integrated Delivery (FARID) project "Health Camp", that delivered basic health and nutrition packages for maternal health, immunisation and nutrition. | From March–December 2023, mobile health teams identified and provided routine immunizations for 51,168 zero dose children <2 years, screened and treated 14,158 malnourished children <5 years, and vaccinated 11,672 pregnant women. Caregivers cited lack of health facilities, safety concerns and opportunity cost as barriers to routine immunisation of their children. Services were disrupted periodically due to security, and in June 2023 a team was attacked and 4 staff held hostage. Services resumed 1 month later. |

*(Continued)*

**Table 1.** (Continued)

| Publication | Geographical region of the intervention | Population | Study design | Sample size | Intervention | Summary of findings relating to children |
|---|---|---|---|---|---|---|
| Kozuki et al (2018) [43] | South Sudan | Children <5 years | Mixed methods | Not reported | Integrated community case management (iCCM) of childhood illness programming in Payinjiar County, Unity State during a period of acute crisis in 2013–2014. | There was a drop in iCCM patient contacts during the crisis, which reached a low point at the height of the crisis and recovered to pre-crisis levels within 3 months. Community health workers continued to provide services throughout the crisis, in spite of some being displaced, worsened security conditions, interruptions in supply and interrupted funding. iCCM contacts were consistently more frequent than health facility contacts for children <5 years even at the height of the crisis. Health workers noted an increase in demand with the influx of IDPs and related increased pressure on supplies. |
| Metzler et al (2019) [49] | Ethiopia | Children and young people residing in refugee camps in Dollo Ado | Mixed methods | Children 6–11 years: 40 attenders and 66 non-attenders Children 12–17 years: 45 attenders and 40 non-attenders | Child and youth learning centres (CYLC) intervention to provide safe spaces for children and youths. Each centre held two three-hour sessions daily: a morning session for children 6–11 years and an afternoon session for youths 12–17 years. The centres included temporary classrooms, latrines for children and staff, an office, storeroom, water tank, kitchen, and a playground with steel-structured play equipment. An informal educational curriculum was delivered by facilitators from the refugee camp and local community, focusing on functional literacy and numeracy skills. Psychosocial activities included cultural dance, drawing, recreational play, and singing. Onsite counselling and a feeding programme were offered. | Children aged 6–11 years who were retained in the programme reported greater needs compared with those lost to follow up (p=0.028). Youth aged 12–17 years who were retained were more likely to report protection concerns compared with youths lost to follow up (p=0.027). Both age groups showed significant improvement in literacy and numeracy from baseline to follow up at 3–6 months. Boys of all ages showed greater improvement in numeracy than girls, and boys 12–17 years had greater improvement in literacy. Psychosocial wellbeing improved for participants as well as non-attenders. Child participants 6–11 years had a larger decrease in difficulties measured using the SDQ compared to non-attenders (p<0.05); the effect was mainly attributed to improvement among boys. Carers of children 6–11 years that did not participate had a greater increase in reporting of protection concerns at follow up compared with caregivers of children who attended the CYLC (p<0.05). CYLC attendance had no significant impact on older children's perceptions of protection risks over time. For older boys, CYLC attendance predicted a greater increase in reporting of protection concerns compared with non-attenders (p<0.05). |
| Poe et al (2024) [82] | Myanmar | Children <1 year | Mixed methods | 184 | Implementation of a routine childhood immunisation programme by a nurse-led civil society organization in an ethnic state during ongoing conflict and an aid embargo that included embargo of vaccines. | Private donations were used to purchase 5 routine childhood vaccines, supplies and cold chain equipment and smuggle these into Myanmar to the programme site. Cold chain standards were met. A census was conducted to identify children <1 year for vaccination. Only 84 (55%) of 153 identified children participated in the vaccination intervention. The parents of 100 additional children that were not identified by the census requested vaccination and were included. 79% of participating children were zero dose and DPT-1 coverage was 18%. 56% of participants received the age-specific recommended doses of BCG, Penta, and OPV. 71% of vaccinated children were IDPs. Challenges included migration of IDPs and vaccine stockouts. |

*(Continued)*

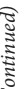

**Table 1.** (Continued)

| Publication | Geographical region of the intervention | Population | Study design | Sample size | Intervention | Summary of findings relating to children |
|---|---|---|---|---|---|---|
| UNICEF (2012) [85] | multiple | Children living in areas affected by armed conflict (global) where a Monitoring and Reporting Mechanism on Grave Violations Against Children exists | Mixed methods | NA | Monitoring and Reporting Mechanism (MRM) on Grave Violations Against Children in situations of Armed Conflict 2005–2012. | The MRM provided a framework for organizing information on violations against children and for concerted efforts to respond to these. In Afghanistan, MRM data led to a mine-risk education project. In the occupied Palestinian Territories, information from the CAAC Working Group database helped identify "vulnerability spots" and led to interventions aimed at reducing harassment and improving access to school. Violations documented by the MRM influenced decisions regarding funding and training of military staff in the DRC. The MRM also helped to identify children in the course of armed groups' integration process into the regular armed forces. In Sri Lanka, MRM data informed targeted prevention and response activities to protect children. The DRC government took steps to stop and prevent the recruitment and use of children in conflict and to protect children from sexual violence based on MRM data. MRM data informed criminal conviction of perpetrators in the DRC. |
| Zachariah et al (2012) [61] | Somalia | Paediatric inpatients | Mixed methods | 3920 paediatric admissions | Daily scheduled telemedicine support for paediatric care in an NGO-run district hospital. Support was given in through audio-visual exchange between clinicians in Somalia and a paediatrician in Nairobi. Criteria were defined for mandatory telemedicine referrals. Education was provided on identification of risk signs, differential diagnoses, medication dosages, choice of antibiotics and follow up requirements. The specialist paediatrician offered counselling to parents of seriously ill children, to foster a positive relationship between the clinical team and parents. | There were 3920 paediatric admissions in 2011, 81% were <5 years. 346 (9%) of paediatric patients were referred for telemedicine. 25% of telemedicine cases included a life-threatening condition that had been initially missed but was diagnosed with telemedicine support. Significant changes to management were made in 64% of cases. The capacity of local clinicians to manage complicated cases improved over time, seen by a linear drop in alterations to initial case management for meningitis and convulsions (92–29%, P=0.001), lower respiratory tract infection (75–47%, P=0.02) and complicated malnutrition (86–40%, P=0.002). Adverse outcomes (deaths and lost to follow up) dropped 30% compared to the preceding year (OR 0.70, P=0.001). There was a concomitant increase in formal referrals to a higher-level care facility (OR 7.6, P=0.02). The number of patients that needed to be treated through telemedicine to prevent one adverse outcome was 45. Staff found telemedicine useful for improving patient care, including improved recognition of risk signs; improved management and prescribing practices; and building solidarity with distant specialist colleagues through direct contact. Telemedicine was accepted by all parents of children; there were no consultation refusals. Disruptions in audio and video quality led to cancellation of 2 sessions. The problem was solved by increasing the internet bandwidth. |

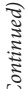

*(Continued)*

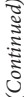

(Continued)

| Publication | Geographical region of the intervention | Population | Study design | Sample size | Intervention | Summary of findings relating to children |
|---|---|---|---|---|---|---|
| Callaghan et al (2024) [79] | Bangladesh | Rohingya refugee children 5–12 years old | Observational | 152 | A 10-day prosocial behaviour intervention was designed in collaboration with Rohingya partners. Children were paired with age- and gender-matched peers who were not related or friends prior to the study. Children engaged in daily sessions in groups of 8, supported by three trained Rohingya field researchers who lived in the camp. Peer pairings engaged collaboratively in play-based activities that targeted cognitive-affective processes involved in prosocial behaviour, including helping, sharing, empathic responding, emotion perspective-taking, executive functioning (behavioural shifting and inhibition). Cognitive affective processes were both a component of the intervention and an outcome measurement. The intervention included certain tasks to promote prosocial behaviour and different tasks to measure outcomes. | Age and birthplace were significant predictors of baseline prosocial behaviour and intervention-related change. After the intervention, children 9–12 years old were more likely to choose equality over inequality during a Forced Choice sharing task. Camp-born children showed improved "how-to" helping behaviours after the intervention, while children born in Myanmar were more likely to take over the task and thus provide "do-for" help. Myanmar-born demonstrated increased sharing and generosity after the intervention, and were more likely than camp-born children to have increased empathic responding such as imagining how they would feel in a situation. Myanmar-born children showed less improvement on executive functioning compared with camp-born children. |
| de Lima Pereira (2018) [63] | Syria | Children 6–59 months | Observational | 3410 | Measles and polio vaccination campaign in a location where active conflict had recently subsided, displaced families began returning, and routine vaccination was not functioning. The study reported on measles only. | At pre-intervention survey, only 79.3% of survey participants had received one vaccine or more in their lifetime and only 20.3% had received all vaccines due by their age. Vaccination coverage was lowest for children <1 year old. A total of 3410 children aged 6–59 months were vaccinated for measles. Post-intervention measles vaccination coverage was 81.8% (95%CI: 76.9-85.9%). Reasons cited for non-vaccination included caretaker was busy during the campaign, lack of knowledge about the vaccination campaign and that the child was sick during the vaccination campaign. |
| Dozio et al (2024) [38] | Central African Republic | Children 6–17 years old with post-traumatic stress | Observational | 661 | Comparison of five bi-weekly group sessions with narrative therapy vs an Eye Movement Desensitization and Reprocessing-based intervention (Group Trauma Episode Protocol (G-TEP)) for children with post-traumatic stress symptoms. During narrative therapy sessions, a theme was discussed (e.g., loss, traumatic events, the future) for 90–120 minutes and participants were invited to share feelings, experiences, and thoughts on their future. Psychoeducation and relaxation and stress management techniques were provided, and a psychosocial worker was present to provide emotional support. The G-TEP group sessions focused on connection to past, present, and future resources, exposure to traumatic memories and alternating bilateral stimulations. G-TEP facilitators were paraprofessionals trained in the interventions. | Both interventions significantly reduced post-traumatic stress scores at post-treatment evaluation (p<0.001 for both measures in both interventions). There was no significant difference in effectiveness between the two types of treatment. Five-month follow up was restricted to 185 children due to security concerns. Five-month follow up showed sustained effect of both interventions. Participants in the G-TEP protocol were significantly older than those in the narrative therapy group. |

| Publication | Geographical region of the intervention | Population | Study design | Sample size | Intervention | Summary of findings relating to children |
|---|---|---|---|---|---|---|
| Haar et al (2020) [81] | Afghanistan | Female caregivers and children aged 8–12 years | Observational | 72 families | Strong Families"intervention, a family skills training programme comprised of 1–2 hour sessions for 3 weeks. Parallel sessions for children and their primary caregiver were followed by joint family sessions. Caregiver sessions explored challenges faced by caregivers, stress management, showing affection whilst setting boundaries, listening to children, encouraging good behaviour and discouraging misbehaviour. Child-specific sessions focused on coping with stress, exploring rules and responsibilities, reflecting on their goals, and reflecting on the roles their caregivers play in their lives. Family sessions focused on positive communication, stress relief techniques, family values and sharing appreciation.. Participants were recruited through schools and drug treatment centres. Facilitators included a mix of teachers, caregivers who had previously taken part in a family skills programme, psychologists and social workers. | Child participants' total difficulty score and all sub-scores of the Strengths and Difficulties Questionnaire (SDQ) reduced significantly at 6-week follow up (p<0.001), with no significant differences by gender. SDQ were similar for children recruited through drug treatment centres and high schools. Parenting practices were evaluated using the Parenting and Family Adjustment Scales (PAFAS). PAFAS scores improved in all four subscales at 6-week follow up, with the greatest improvements in parents who had higher scores at baseline. Family adjustment scores also improved significantly at follow up with greater improvement for caregivers with higher scores at baseline. |
| Hermosilla et al (2019) [85] | multiple | Children 6–17 years old | Meta analysis of observational studies | 1010 children and 1312 carers | Child Friendly Spaces (CFS) interventions in 5 countries. Sites in Ethiopia, Uganda, Iraq, and Jordan supported conflict-affected children. The Nepalese intervention was post-earthquake. Activities included support for functional literacy and numeracy skills (Ethiopia, data from January – May 2012); traditional song, dance, storytelling, and organized sport (Uganda, data from October 2012 – March 2013); music, sports, drawing, storytelling, drama, and dance (Iraq, data from September 2013 - March 2014); drawing, handicrafts, puzzles, games, storytelling, singing, and drama (Jordan, data from February - August 2014), and games, outdoor sports, creative activities, traditional song and dance, and life skills activities (Nepal, data from March 2015 - May 2016). | In the younger age group (6–11 years), significant intervention impacts were observed for protection concerns in Ethiopia (Cohen's d = 0.48, 95% CI 0.08–0.88), psychosocial wellbeing in Ethiopia (d = 0.51, 95% CI 0.10–0.91) and Uganda (d = 0.21, 95% CI 0.02–0.40), and for developmental assets in CFS attenders in Uganda (d = 0.37, 95% CI 0.15–0.59) and Iraq (d = 0.86, 95% CI 0.18–1.54). Pooled analyses for children 6–11 years showed statistically significant improvement in CFS attenders for psychosocial wellbeing only (d = 0.18, 95% CI 0.03–0.33). Among children aged 12–17 years, attending the CFS in Iraq was associated with reduced reporting of protection concerns (d = 0.58, 95% CI 0.07–1.09), however the effect disappeared in pooled analysis. Site-specific and pooled analyses for all other measures were not significant in the older age group. |
| Kohrt et al (2015) [82] | Nepal | Children associated with armed forces and armed groups | Observational | 258 | Comparison of education-based reintegration programmes for children associated with armed forces and armed groups vs no reintegration support. Programme options included return to school, vocational training, apprenticeship, or income generation support via micro-grants or loans. Children selected the intervention of their choice, and they participated in only one type of reintegration programme. The packages were implemented by local NGOs. All children were offered post-war psychosocial support programmes together with other children the community. | Children receiving the education reintegration packages were less likely to be female or from lower castes and were more likely to be younger compared with children in the other treatment groups and the no treatment group. Findings were adjusted using propensity score weighting. Compared with no treatment, children participating in education reintegration packages and other reintegration packages had a non-significant reductions in depression scores and nonsignificant increases in post-traumatic stress and functional impairment. |

*(Continued)*

**Table 1.** (Continued)

| Publication | Geographical region of the intervention | Population | Study design | Sample size | Intervention | Summary of findings relating to children |
|---|---|---|---|---|---|---|
| Leidman et al (2017) [45] | Nigeria | Children <5 years | Observational | 7791 | Four types of interventions, including 1) a measles vaccination campaign, 2) a public health outreach campaign, 3) anthelmintic prophylaxis, and 4) distribution of fortified cereals. Health outcomes, health service access, service quality and mortality were estimated using a series of two-stage cluster surveys. | Global acute malnutrition in children aged 0–59 months decreased significantly in the second survey. Crude and <5 mortality rates increased in all regions, exceeding emergency thresholds in Central Borno, and Central and Northern and Southern Yobe. Measles vaccination coverage increased significantly in all regions except Southern Borno. In Central Borno and Northern Yobe, measles vaccination coverage doubled. None of the included regions attained 95% vaccination coverage for herd immunity. Households reported significantly improved access to maternal, neonatal, and child health services (e.g., vaccinations, nutritional screening, birth registration, and bed net distributions) and significantly improved anthelmintic prophylaxis use. There was no change in the reported distributions of fortified cereals. Diarrhoea prevalence increased significantly in all Borno regions and was not significantly changed in Yobe region after the delivery of the humanitarian interventions. Less than one-third of children with diarrhoea received oral rehydration solution, zinc, or both in the baseline survey or follow up survey, and <8% received appropriate treatment. In Southern Yobe and Central Yobe, the proportion of children receiving zinc significantly declined after the interventions. |
| Mashal et al (2024) [49] | Sudan | Children <5 years | Observational | N/A | Vaccination campaign in response to vaccine-derived polio outbreak | Two rounds of immunisation activities were undertaken using mOPV2 for children 0–5 years of age. The first round in November 2020 reached 97% of the targeted 8.5 million children <5 years. The second round January - February 2021 reached 98% of the targeted 8.5 million children <5 years of age according to an independent monitoring survey of 69,279 households. Administrative data showed all localities above 79% coverage and most were above 95%. The second round included co-administration of Vitamin A, targeting 7.6 million children and achieving 99% coverage. |
| Koris et al (2022) [43] | Nigeria | Adolescent girls 10–14 years, their brothers 15–19 years, and male and female caregivers | Qualitative | 86 | The Sibling Support for Adolescents in Emergencies (SSAGE) program, a whole-family support intervention to build adolescent girls' protective assets against violence, challenge intergenerational cycles of violence and prevent future GBV. Participants included adolescent girls, their male siblings, and their male and female caregivers. Four distinct curricula focused on topics of gender, power, violence, interpersonal communication, and healthy relationships. Participants engaged in synchronized interactive age and gender-specific sessions. Intra-familial discussion on weekly topics was encouraged. The SSAGE curricula were adapted from Mercy Corps Nigeria's Life Skills curriculum, through a community-based participatory design process with IDP community representatives. | One month after completing the programme, participants reported increased understanding of gendered power differentials, improvement in communication amongst family members, changes in parenting style, and reduced violence and greater harmony within households. Changes in male caregivers' communication style appeared to have a broadly positive impact on the entire household. Male siblings also reported changes in attitudes and expectations towards their sisters, with reduced conflict and violence, and increased empathy. However, norms of patriarchal dominance were maintained, and some participants ascribed improved harmony in the household to improved obedience. Participants endorsed adolescent girls' right to protection from violence in the community and improved awareness of services to protect and treat survivors of violence. |

*(Continued)*

| Publication | Geographical region of the intervention | Population | Study design | Sample size | Intervention | Summary of findings relating to children |
|---|---|---|---|---|---|---|
| Lilleston et al (2018) [68] | Lebanon | Syrian refugee and host population women and girls ≥14 years | Qualitative | 75 | GBV services delivered by mobile health teams comprised of 3 female staff: a community mobilizer, a caseworker and an adolescent girls' assistant, and a male community mobilizer who rotated between the three teams. The intervention included psychosocial support activities (e.g., support groups and craft activities), risk mitigation activities (e.g., safety planning, service mapping), and individual case management. The intervention activities at each site were selected by participants and timed to optimise access. Female focal points were identified from the Syrian refugee population to support community engagement, dissemination of information, and refer GBV survivors. Services were delivered in spaces identified by the community as familiar, safe and comfortable for women and girls. Services ranged from short-term emergency relief to a 6-month expanded service package. | Most of the study findings were not disaggregated by age. Adolescent participants reported improved social connectedness and development of supportive relationships as a result of attending services. Adolescents also reported improved emotional support from the programme activities and programme staff, improved emotional regulation, and increased knowledge about mental health, coping and communication techniques. Adolescents reported an increased sense of safety and confidence in public spaces. Adult caregivers reported improved mental health, increased knowledge about parenting and reduced use of corporal punishment. |
| Schafer et al (2016) [73] | occupied Palestinian Territories | Families living in Gaza | Qualitative | 38 (27 adults and 11 children) | Psychological First Aid (PFA) after Operation Protective Edge in July-August 2014. 300 individuals trained in PFA offered services to 13,400 households, reaching approximately 61,000 individuals. Participants were reached via door-to-door home visits to enquire about mental health support needs, referrals from community members, and for groups for mothers with children attending child friendly space programs. | PFA was not considered sufficient to meet the needs of the affected population. Participants stated that other psychosocial support approaches were needed and that PFA should be directly linked with material supports. PFA was reported to contribute to improved short-term and long-term functioning, including impacts on how parents supported their children during stressful times. The benefits of PFA included reduced distress, facilitation of physical safety of participants, increased calming strategies for caregivers and children, fostering of social connections, improved sense of control, and promotion of hopefulness. Men were more likely to engage in PFA when it was delivered in the home, to the entire family. Child FGD participants did not distinguish PFA as a separate approach from the general psychosocial support they received within CFS program activities. Nonspecialists providing PFA interpreted "do no harm" in terms of safety and protection, and had limited awareness of how the provision of psychosocial services can cause harm. PFA trainers reported instances where PFA providers pushed people to speak about their experiences. |
| Ahmadi et al (2022) [77] | Afghanistan | Hazara girls aged 12–18 years who experienced a school terrorist attack, and displayed heightened PTSD symptoms | Randomized controlled trial | 120 | Modified written exposure therapy (m-WET) was compared with intensive trauma-focused group cognitive behaviour therapy (TF-CBT) and a control group with no intervention. The m-WET group participated in 5 daily sessions of psycho-education and writing exercises in groups of 5–8 adolescents, focusing on the terror attack. The TF-CBT group included 5 group sessions with a clinical psychologist, with specialist training in TF-CBT. | At the end of the intervention, the m-WET and TF-CBT groups had significantly lower PTSD symptom severity than the control group ($p < 0.001$ and $p < 0.01$, respectively). The TF-CBT and m-WET groups did not differ significantly. These effects were maintained at 3 month follow up. Overall satisfaction with m-WET was high. There were no unintended or harmful effects in either treatment group. |

*(Continued)*

**Table 1.** (Continued)

| Publication | Geographical region of the intervention | Population | Study design | Sample size | Intervention | Summary of findings relating to children |
|---|---|---|---|---|---|---|
| Ahmadi et al (2023) [75] | Afghanistan | Girls aged 11–19 years with psychiatric distress | Randomized controlled trial | 125 | Memory Training for Recovery–Adolescent (METRA), a 10-session group intervention delivered by facilitators with a health or education background and minimal training. The intervention focused on memory specificity and trauma writing. The control group received "treatment as usual" delivered by a local NGO, which included 10 group sessions on mental health literacy, relationships, puberty, and maintaining physical and mental health. Interventions were delivered in a rented property over 2 weeks because girls were excluded from school. | The intervention group had greater reduction in PTSD ($p<0.001$) and Afghan-cultural distress symptoms compared with controls ($p<0.001$). The intervention group also had decreases in depression, anxiety, and psychiatric difficulties from baseline, while the control group had increases ($p<0.001$ for all three measures). At 3 month follow up, the intervention group had sustained reduction in PTSD, depression, and anxiety symptoms, psychiatric difficulties, and Afghan-cultural distress symptoms ($p \le 0.001$ for all measures). Seven participants were excluded during the intervention and referred for psychiatric care. Qualitative feedback from study participants was positive. |
| Ahmadi et al (2023) [75] | Afghanistan | Boys aged 14–19 years with PTSD symptoms | Pilot randomised controlled trial | 58 | School based intervention: Memory Training for Recovery–Adolescent (METRA), a 10-session group intervention that was delivered over 2 weeks by counsellors with mental health training. The intervention focused on memory specificity and trauma writing. The control group received 10 group sessions on creative and critical thinking, study methods, and problem-solving skills. | The intervention group had reduction in PTSD symptoms ($p<0.001$) from baseline and reduced depression, anxiety, psychiatric difficulties, and Afghan-cultural distress symptoms ($p<0.01$ for all measures). The control group had milder decreases in PTSD, depression, and Afghan-cultural distress symptoms, and slight increases in anxiety symptoms and psychological distress. At 3-month follow up, intervention group symptom reductions were sustained and significant compared to controls ($p=0.03$ for depression symptoms and $p \le 0.01$ for all other measures). Nearly half of participants in both groups dropped out; reasons cited were interference of the programme with class time and teachers not supportive of attendance. Participants reported that participation in the mental health study could be seen as weakness and led to feelings of shame. |
| Bryant et al (2022) [62] | Jordan | Syrian refugees aged 10–14 years with psychological distress | Single-blind, parallel, controlled trial | 471 | Early Adolescent Skills for Emotions (EASE), comprised of 7 weekly 1.5-hour sessions for gender-specific groups of 8–10 adolescents. The curriculum included psychoeducation, coping strategies, behavioural activation, problem solving, and relapse prevention. One caregiver of each adolescent was invited to three 2-hour group sessions that promoted supportive parenting. The trained facilitators had a range of professional backgrounds but were not mental health specialists. Participants in the control group received "enhanced usual care (EUC)" through a single 30-minute family session conducted in the participant's home by a community health worker, where they received feedback on the adolescent's assessment, information on coping strategies, and a list of local services available for psychosocial support. | At 3-month follow up, the intervention group had greater reductions in internalising symptoms ($p=0.007$), psychological distress ($p=0.002$), and inconsistent disciplinary parenting ($p=0.01$) compared with controls. There were no differences in externalising or attentional symptoms, or secondary outcomes of depression, posttraumatic stress, well-being, functioning, school belonging, and caregivers' parenting and mental health. There were no differences in parental involvement, positive parenting, poor supervision and monitoring, or corporal punishment. Reductions in internalising symptoms in the intervention group were mediated by reductions in inconsistent disciplinary behaviours by caregivers. Six severe adverse events were reported during the trial, 2 in the intervention group and 4 in the control group, involving suicidal risk in adolescents or caregivers, or threat of violence to adolescents. None of the adverse events were related to the interventions. |

*(Continued)*

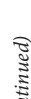 PLOS Global Public Health

**Table 1.** (Continued)

| Publication | Geographical region of the intervention | Population | Study design | Sample size | Intervention | Summary of findings relating to children |
|---|---|---|---|---|---|---|
| El-Khani et al (2019) [65] | occupied Palestinian Territories | Caregivers of children 8–14 years old | Waitlist randomised controlled trial | 120 | "Caregiving for Children through Conflict and Displacement", a single 2-hour session focused on building parenting confidence and self-regulatory skills and improving child and family psychological well-being. The intervention was delivered by two facilitators to groups of up to 20 caregivers. A booklet about parenting skills was provided to caregivers covering topics including how to provide warmth and support, maintenance of routines, encouraging play, giving praise, relaxation techniques, and managing aggressive behaviour. A presentation that complements the booklet was delivered, to facilitate discourse and questions among caregivers. Facilitators worked for a supporting local NGO and self-trained using a facilitator guide. | The study was primarily focused on feasibility but also measured MHPSS outcomes. The intervention was feasible for nonspecialist facilitators to deliver and feasible to evaluate. At three-month follow up, participants had significantly improved parenting and family adjustment scores compared with the control group. Participants reported significantly improved measures of emotional problems, hyperactivity/inattention, pro-social behaviour and total difficulties scores at 3-month follow up. There was a nonsignificant improvement in parenting and reduction in child emotional and behavioural difficulties in the control group at follow up compared to baseline. |
| El-Khani et al 2021 [64] | Lebanon | Refugee families with a child 9–12 years who with post-traumatic stress | Randomised controlled trial | 119 | An enhanced Teaching Recovery Techniques (TRT) intervention was delivered, expanding the caregiver sessions. The intervention included five 2-hour group sessions with children focused on memories, nightmares, flashbacks, relaxation problems, concentration, sleep, and fears associated with war. In addition to the standard adult caregiver sessions on normalising children's reactions and caregiver coping, an additional three caregiver sessions were added, focusing on evidence-based parenting strategies to build self-regulatory skills and reduce child emotional and behavioural difficulties. The intervention was delivered by teachers trained in TRT and the additional three caregiver sessions. | Both treatment groups (enhanced TRT group and standard TRT) had significant improvements. The enhanced TRT group had a significantly higher intrusion scores at baseline compared to the standard TRT group (p=0.042) and the waitlist group (p=0.015). All three groups had significantly improved intrusion scores at 3 month follow up, with the two treatment groups showing greater improvement than controls (p<0.001). Both treatment groups had significantly improved avoidance, arousal, depression self-rating, and both child- and caregiver-reported anxiety scores compared with waitlist controls. The enhanced TRT group showed greater improvement in avoidance (p=0.019). There were no significant changes in Strengths and Difficulties Questionnaire scoring for all groups at 3 month follow up. Caregiver stress scores and post-traumatic stress scores improved significantly for all three groups at 3 month follow up. The two treatment groups showed significantly greater improvement in post-traumatic stress compared with waitlist controls (p<0.001). Both treatment groups had significant improvement in anxiety and depression scores compared with waitlist controls. Parenting skills scores improved significantly for both treatment groups over time (p<0.001 for enhanced TRT and p=0.001 for standard TRT). There were no treatment effects on parenting confidence scores. |
| Fabbri et al (2021) [40] | Tanzania | Children ≥9 years attending schools in the refugee camp | Cluster randomised controlled trial | 4729 | The EmpaTeach intervention, a 10 week peer-led intervention focused on reducing impulsive violence by teachers against school children. The intervention used empathy-building exercises and group work to improve teachers' self-efficacy, self-regulation, and empathy for students, and to reduce teachers' stress levels and create social support for skill development. Information was provided on alternative disciplinary methods and positive classroom management strategies. | There was no evidence that the intervention reduced physical violence or emotional violence by teachers towards primary or secondary school students in the study site. There was no effect on students' depressive symptoms or school attendance. Children in intervention and control arms reported high levels of violence in school: In the baseline, midline and endline surveys, 47.8-53.9% reported experience of physical violence and 15.4-18.5% reported emotional violence from a teacher during the previous week. The trial was impacted by a mass layoff of 20% of teachers in the camp during the intervention delivery period. Most but not all teachers were re-hired in the same role in the same school within 2 weeks. |

*(Continued)*

**Table 1.** (Continued)

| Publication | Geographical region of the intervention | Population | Study design | Sample size | Intervention | Summary of findings relating to children |
|---|---|---|---|---|---|---|
| Hermosilla et al (2022) [67] | Jordan | Syrian and Palestinian refugees and host population aged 6–17 years | Waitlist controlled trial | 406 | Child friendly space for children 5–12 years and 13–17 years. Separate programmes were offered to the two age groups, consisting of 2-hour sessions three days per week for up to 12 weeks. The intervention included structured creative and play activities, informational videos, and sessions on life skills with topics including hygiene, community mapping, and volunteerism. | Participation in the Child Friendly Space did not predict mental health, child protection concerns, caregiver stress, or developmental assets outcomes 15 months after the baseline assessment. There were no statistically significant differences in school attendance between those who participated in the Child Friendly Space and controls. |
| Metzler et al (2023) [51] | Uganda | Adolescents 9–14 years old | Randomized controlled trial | 795 | A structured psychosocial intervention to support mental health, development, and protection. The 12-week program was implemented in Child Friendly Spaces, and comprised of 3-hour daily sessions focused on community building, emotional learning, wellbeing and coping, social support, relating to others, protection and boundaries, and building on strengths. The intervention was developed in consultation with adolescents, adapted and pretested in a neighbouring South Sudanese refugee response. Outcomes were compared with standard treatment and waitlist controls. | Only 15.7% of study intervention participants and 21.1% of standard treatment participants attended ≥50% of sessions. At 12 months after baseline, psychological distress scores decreased, and resilience scores increased for all three groups. The intervention and standard approaches were more effective in reducing psychological distress and perceived protection risks compared with no intervention. Compared to standard treatment, the intervention group had greater improvement in relational scores (p=0.025) and overall resilience (p=0.033). There was no significant difference between the intervention and standard treatment groups on post-traumatic stress. All groups improved in development and hope scores, functional literacy and numeracy, and positive coping strategies. The intervention group had greater improvement in developmental assets compared with waitlist controls (p=0.013). Protection risks reduced in the intervention and standard treatment groups compared to controls, but negative coping strategies increased over time in all three groups. |
| Miller KE et al (2022) [69] | Lebanon | Refugees and host population adult caregivers with at least one child between 3–12 years old | Randomised controlled trial | 480 caregivers from 240 families | A caregiver support intervention was delivered in 9 weekly sessions for primary caregivers of children aged 3–12 years affected by armed conflict and forced migration. The intervention was facilitated by Syrian, Lebanese, and Palestinian non-specialists, with an equal number of women and men. The intervention focused on improving parenting knowledge and skills and improving the mental health and psychosocial wellbeing of participants. Sessions focused on caregiver wellbeing, stress management techniques, strengthening parenting in adversity, increasing positive parent–child interactions, and decreasing the use of harsh parenting practices. | The trial was adapted for the second wave of the intervention due to the Covid-19 pandemic. There was no significant change on overall parenting skills at endline or follow up. In the sub-sample that received the full intervention, overall parenting skills improved (d=0.25, p<0.05). There was significant improvement in the full sample at endline and follow up for measures of harsh parenting (p<0.05), parenting knowledge (p<0.001), and caregiver distress (endline p<0.001; follow up p<0.01). The changes in harsh parenting were partly mediated by improvement in caregiver wellbeing. There were no significant changes in parental warmth and responsiveness, psychosocial wellbeing, stress, or stress management. |

*(Continued)*

**Table 1.** (Continued)

| Publication | Geographical region of the intervention | Population | Study design | Sample size | Intervention | Summary of findings relating to children |
|---|---|---|---|---|---|---|
| Panter-Brick et al (2018) [70] | Northern Jordan | Refugee and host population youth 12–18 years old | Randomised controlled trial | 817 | The Advancing Adolescents programme focused on providing safety, support, and group-based activities in Adolescent Friendly Spaces in 16 sessions over 8 weeks. Groups were gender differentiated, with a mix of 10–15 youth Syrian refugees and Jordanian youth. Sessions were run by trained adult lay facilitators from the local community. Youth chose from a range of programme modalities that included fitness, arts/crafts, vocational skills and technical skills. Transport to the programme was provided, where needed. | Syrian refugees had experienced significantly more traumatic events than Jordanians (p <0.0001) and most often reported trauma relating to armed conflict. Jordanian youth described witnessing violence, barriers in access to medical care, or being in an accident. 34.8% of study participants were lost to follow up. The intervention improved human insecurity, distress, perceived stress, and mental health difficulties, with significant medium to small effect sizes. There was no effect on prosocial behaviour or post-traumatic stress. Trauma exposure predicted endline score, with stronger beneficial impacts of the intervention in the high-trauma versus low-trauma cohort. Technical and vocational skills activities had stronger effects relative to fitness activities or arts and crafts. Age and gender did not impact outcomes. Symptom scores improved for both treatment and control groups over time, with an overall trend toward recovery. There were sustained positive impacts on human insecurity for the intervention group relative to the control group after 7–14 months. |
| Panter-Brick et al (2020) [71] | Northern Jordan | Refugee and host population youth 12–18 years old | Randomised controlled trial | 817 | The Advancing Adolescents programme focused on providing safety, support, and group-based activities in Adolescent Friendly Spaces in 16 sessions over 8 weeks. Groups were gender differentiated, with a mix of 10–15 youth Syrian refugees and Jordanian youth. Sessions were run by trained adult lay facilitators from the local community. Youth chose from a range of programme modalities that included fitness, arts/crafts, vocational skills and technical skills. Transport to the programme was provided, where needed. | There were three distinct trajectories of inflammatory markers (high, rising, and low c-reactive protein (CRP)), two for cell-mediated immunity (high and low Epstein Barr Virus (EBV) titres), and three for hair cortisol (HCC) (hyper, medium, and hyposecretion). Body mass index was associated with CRP and HCC. CRP trajectories were associated with levels of perceived stress. The HCC trajectories were associated with perceived insecurity. The intervention had a positive effect on cortisol regulation. Participants receiving the intervention showed attenuated cortisol production, with reduced 37.7% reduction in HCC levels over 11 months (p=0.005). The intervention had no detectable impact on CRP or EBV levels. There was no evidence for moderation by gender or trauma exposure. |
| Pluess et al (2024) [72] | Lebanon | Syrian refugee children 8–17 years old with a common mental disorder | Single blind pilot randomised controlled trial | 20 | A telephone-based psychotherapy intervention (t-CETA) was developed based on the Common Elements Treatment Approach (CETA). Children were randomised to receive t-CETA or treatment as usual with standard case management and psychotherapy in clinics from an iNGO. All patients met diagnostic criteria for at least one common mental disorder including depression, any category of anxiety disorder, post-traumatic stress disorder, or conduct or oppositional defiant disorder. t-CETA was delivered by two trained lay counsellors under the supervision of a local psychotherapist who was supervised by a CETA expert. | Participants' diagnoses included major depressive disorder; PTSD; a range of anxiety disorders, obsessive compulsive disorder; and conduct or oppositional defiant disorder. Most children met criteria for more than one disorder. t-CETA was associated with clinically relevant reductions in emotional and behavioural problems compared with controls. There were no differences between the treatment and control groups for global disability, the individual mental health symptoms scales, or well-being scales. 90% of the t-CETA group started the intervention and 60% completed the full course of treatment. 60% of controls started treatment and none completed it. |

*(Continued)*

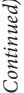

**Table 1.** (Continued)

| Publication | Geographical region of the intervention | Population | Study design | Sample size | Intervention | Summary of findings relating to children |
|---|---|---|---|---|---|---|
| Seal et al (2023) [55] | Somalia | Internally displaced children <5 years old | Cluster randomized controlled trial | 1269 | A Participatory Learning and Action cycle to increase routine childhood immunisation uptake was adapted in partnership with preexisting traditional social groups for adult females. The intervention consisted of 8 weekly group meetings led by two trained facilitators. The groups identified and addressed child health and vaccination topics, analysed challenges, and planned and implemented potential solutions. A partner NGO provided vaccinations to children through mobile clinics in the study area. | Caregiver knowledge in the intervention group increased significantly at endline (aOR 7.89, p<0.0001). Educational status, having received a vaccination, and age were correlated with a higher maternal knowledge scores at baseline and endline. The intervention significantly improved first dose measles vaccination coverage (aOR 2.43, p<0.0001) and pentavalent series completion (aOR 2.45; p=0.008). There was no effect on timeliness of vaccination. The intervention group had a significant increase in possession of a child health card at endline (aOR 2.86; p=0.006). |
| Seff et al (2021) [56] | Democratic Republic of the Congo | Girls 10–14 years old and their caregivers | Mixed-methods, cluster-randomised controlled trial | 732 girl - caregiver dyads | Creating Opportunities through Mentorship, Parental Involvement, and Safe Spaces (COMPASS) programme for adolescent girls aged 10–14 years and a caregiver of their choosing. The intervention was a 12-month programme that included the provision of safe spaces, life skills and social assets building, mentorship of girls, and engaging caregivers as support systems and advocates for girls. | Increases in caregivers' gender-equitable attitude scores were associated with greater likelihood for girls to have increased school participation (aOR 1.08; 95% CI [1.005, 1.154]; p=0.036). There was no association between changes in caregivers' attitudes and girls' experience of sexual or emotional violence or feeling uncared for. Older girls were less likely to experience improved school participation (aOR 0.77; 95% CI [0.686, 0.861]; p<0.001), reduced physical violence (aOR 0.86; 95% CI [0.757, 0.984]; p=0.028), or reduced emotional violence (aOR 0.82; 95% CI [0.717, 0.942]; p=0.005). |
| Slone et al (2013) [74] | Israel | Jewish highschoolers aged 16–17 years | Randomised controlled trial | 179 | School-based social support and self-efficacy intervention in Ashkelon, Israel after Operation Cast Lead with grade 10 students. Classroom-based sessions were implemented twice weekly for 6 weeks immediately upon the return to school after the ceasefire declaration. The intervention included activities, discussions, poetry readings, stories and viewing of movie clips of popular Hebrew songs dealing with self-efficacy and social support. A control group received an intervention with the same format but focused on general issues relating to adolescence. | The intervention group reported increased ability to mobilize support (p<0.0001) and decreased psychological distress (p<0.0001) after completing the programme, as well as increased self-efficacy compared with controls (p=0.03). The intervention was associated with reduced emotional symptoms in (p<0.0001). There were no significant differences between intervention and control groups in changes in the total behavioural difficulties measure. The control group reported decreased ability to mobilise support (p<0.0001) and increased psychological distress (p<0.0001). |
| Stark et al (2018) [57] | Democratic Republic of the Congo | Girls 10–14 years old and their caregivers | Mixed-methods, cluster-randomized controlled trial | 869 adolescent girls aged 10–14 and 764 caregivers | Creating Opportunities through Mentorship, Parental Involvement, and Safe Spaces (COMPASS) programme for adolescent girls aged 10–14 years and a caregiver of their choosing. The intervention was a 12-month programme that included the provision of safe spaces, life skills and social assets building, mentorship of girls, and engaging caregivers as support systems and advocates for girls. | Inclusion of a caregiver component to the COMPASS life skills programme did not reduce participants' experience of any form of sexual violence, physical violence, neglect, child marriage, or transactional sex for adolescent girls. Both intervention and control groups reported a decrease in experience of violence between baseline and endline. The intervention was associated with greater warmth and affection in parenting and reduced rejection compared with controls, for parents who adhered to the programme protocol; caregivers in the treatment arm who did not adhere to protocol were not more likely than those in the wait-list control arm to demonstrate greater warmth and affection or lower overall rejection at the endline. The intervention did not influence caregivers' attitudes toward gender inequitable norms or acceptance of physical discipline. |

*(Continued)*

**Table 1.** (Continued)

| Publication | Geographical region of the intervention | Population | Study design | Sample size | Intervention | Summary of findings relating to children |
|---|---|---|---|---|---|---|
| Stark et al (2018) [57] | Ethiopia | Sudanese and South Sudanese refugee girls 13–19 years old and their caregivers | Waitlist, cluster randomised controlled trial | 919 girls | Creating Opportunities through Mentorship, Parental Involvement, and Safe Spaces (COM-PASS) programme for adolescent girls aged 10–14 years and a caregiver of their choosing. The intervention was a 12-month programme that included the provision of safe spaces, life skills and social assets building, mentorship of girls, and engaging caregivers as support systems and advocates for girls. | Both the intervention group and waitlist controls had high levels of exposure to all forms of violence in the preceding year at baseline. The intervention did not have significant effects on self-reported experience of sexual violence, physical violence, emotional violence, transactional sex, or perceived feelings of safety. Girls who were married or living with someone as if married at baseline were less likely to be married at endline compared with controls (OR 0.57; 95% CI 0.34 to 0.95, p=0.032). The intervention was associated with changes in gender attitudes and improved social support networks. Participants had more supportive attitudes towards schooling (β= 1.08, 95% CI 0.44 to 1.761), p=0.001), delaying marriage (aOR=1.88, 95% CI 1.07 to 3.28, p=0.027), and delaying childbearing until after 18 years of age (aOR=2.04, 95% CI 1.25 to 3.34, p=0.005). The intervention group were more likely to have friends their own age at endline (aOR 1.71, 95% CI (1.18 to 2.49, p=0.005) and have a trusted non-family female adult in their life (aOR 1.997, 95% CI 1.44 to 2.76, p<0.001). |
| Stark et al (2024) [58] | Uganda | Refugees caregivers ≥18 years old | Waitlist-control quasi-experimental design | 1137 | The adapted "Journey of Life" intervention, a parenting programme for caregivers, was comprised of 12 weekly 2-hour group sessions on psychoeducation, self-care, positive parenting, understanding children's needs, identifying children in need of help, building on their strengths, developing a community action plan, and problem-solving. | After adjusting for baseline demographics and all outcome, intervention participants had significantly improved mental distress, social support, functioning, parental warmth and affection, undifferentiated rejection, and attitudes about violence against children (p<0.001 for all effects). Attending all of the final eight sessions was associated with improved mental distress (p<0.05). |
| Tol et al (2014) [60] | Burundi | Children 8–17 years | Cluster randomised controlled trial | 329 | A classroom-based intervention (CBI) was nested in a larger school-based intervention comprised of structured social activities and mental health treatments. The CBI intervention included 15 sessions over five weeks delivered by non-specialized facilitators. Activities included cognitive behavioural (psychoeducation, coping, and discussion of past traumatic events through drawing) and creative expression activities (cooperative games, structured movement, music, drama, and dance) with groups of around 15 children. | There were no statistically significant differences for the intervention group on outcomes or mediators. Household size and makeup moderated outcomes in the intervention group. Children living in larger households had greater improvements for depressive symptoms and function impairment trajectories. Additionally, living with both parents was associated with statistically significant decreases in PTSD and depressive symptoms in the intervention group. Living with both parents was associated with significant increases in depressive symptoms over time in the waitlist control condition. Age and exposure to traumatic events also moderated outcomes in the intervention group, where younger age and fewer exposures was associated with increased hope. These effects were not seen in the control group. |

**Table 2. Overview of included studies.**

| Characteristic of included studies | | | Number of studies, N (%) | | |
|---|---|---|---|---|---|
| **Study Design** | Trial | | | 20 | (39.2%) |
| | | Randomised controlled trial | 17 | | (33.3%) |
| | | Wait-list controlled trial | 2 | | (3.9%) |
| | | Single-blind parallel controlled trial | 1 | | (2.0%) |
| | Mixed methods | | | 12 | (23.5%) |
| | Observational | Before and after | | 8 | (15.7%) |
| | Cross-sectional | | | 4 | (7.8%) |
| | Case-control | | | 1 | (2%) |
| | Retrospective cohort | | | 1 | (2%) |
| | Qualitative | | | 3 | (5.9%) |
| | Descriptive | | | 2 | (3.9%) |
| | Total | | | 51 | (100%) |
| **Thematic focus** | | | | | |
| | MHPSS | | | 29 | (56.9%) |
| | | Focused, non-specialised services | 20 | | (39.2%) |
| | | Specialised services | 9 | | (17.6%) |
| | Child protection | | | 20 | (39.2%) |
| | | Friendly Spaces or Safe Spaces | 11 | | (21.6%) |
| | Somatic health | | | 19 | (37.3%) |
| | | Immunisation | 8 | | (15.7%) |
| | | Nutrition (included other outcomes) | 5 | | (9.8%) |
| | | Sexual and reproductive health | 3 | | (5.9%) |
| | | Infectious diseases | 2 | | (3.9%) |
| | | Telemedicine for paediatrics | 2 | | (3.9%) |
| | | Mortality | 2 | | (3.9%) |
| | | Toxic stress | 1 | | (2%) |
| | Parenting | | | 15 | (29.4%) |
| | Access to care | | | 9 | (17.6%) |
| | Child development | | | 5 | (9.8%) |
| | Education | | | 3 | (5.9%) |
| **Population displacement status** | | | | | |
| | Not specified | | | 23 | (45.1%) |
| | Refugee | | | 17 | (33.3%) |
| | IDP | | | 11 | (21.6%) |
| | Total | | | 51 | (100%) |
| **Geographic location** | | | | | |
| | Africa | | | 26 | (51%) |
| | Middle East | | | 13 | (25.5%) |
| | Asia | | | 9 | (17.6%) |
| | Multi-centre | | | 2 | (3.9%) |
| | Europe | | | 1 | (2%) |
| | the Americas | | | 0 | (0%) |
| | Total | | | 51 | (100%) |

*Individual studies could include more than one thematic focus

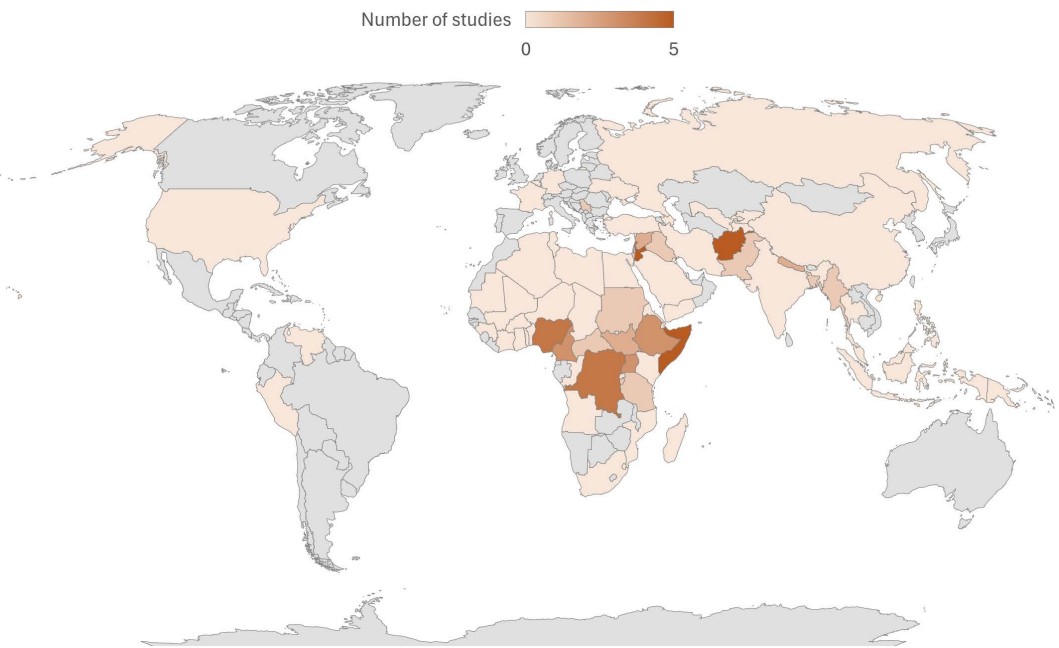

**Fig 2. Geographical distribution of included studies.** Caption: The colour coding applies to entire countries and not regions within countries. All countries that experienced conflict during 2012-2024 are shaded, with darker shading corresponding to the number of studies on child health interventions. This map does not include NATO forces sent from numerous countries to conflicts in other countries. Countries in conflict source: Uppsala Conflict Data Program [86] and the Armed Conflict Location & Event Data Conflict Index 2024 [87].

outcomes in two, or all three of MHPSS, child protection, and parenting support. The MHPSS interventions were primarily group-based interventions (N = 22/29) with widely varying designs, including: modified written exposure therapy [76], memory training techniques [74,75], psychoeducation and support for coping and emotional and behavioural regulation [47,50,59,61,63], remotely delivered psychotherapy [71], Eye Movement Desensitization and Reprocessing-based group therapy [37], narrative therapy [37], social skills building and self-efficacy training [73], Baby Friendly Spaces [38], Child/Adolescent Friendly Spaces or Safe Spaces [49,56,66,69,70,77,84], Psychological First Aid (PFA) [72], reintegration programmes for children associated with armed forces or armed groups [81], mobile gender-based violence (GBV) services [67], a microfinance programme [40], and interventions for caregivers focused on parenting skills and psychoeducation [57,63,64,68,79,80,83]. Remarkably, 25 of the 29 interventions reported clear positive mental health and/or psychosocial support outcomes amongst children participating in the intervention. Of these, only one third (N = 8/25) of interventions were delivered by staff with mental health expertise [37,38,57,67,73–75,80]; the remaining 17 interventions were delivered by non-specialists [40,49,50,56,61,63,64,68–72,76,77,79,83,84].

A series of small studies explored rapid, light-touch caregiver support group sessions focused on parenting skills and emotional self-regulation [64,80,83]. One programme in Palestine involved a single 2-hour parenting and emotional self-regulation information session delivered by non-specialists to 120 caregivers [64]. At three month follow up, participating caregivers reported significantly improved parenting and family adjustment scores, as well as improved symptoms of emotional problems, reduced hyperactivity/inattention, improved prosocial behaviour and reduction in the total difficulties scores in their children when compared to the control group. Two studies evaluated a family skills programme with parallel and joint sessions for caregivers and children 8–15 years old, delivered during three weekly sessions lasting 1–2 hours each (five hours total) [80,83]. The studies evaluated outcomes in Afghanistan (N = 72) and Afghan refugee families living in reception centres in Serbia (N = 25), respectively. At 6-week follow up, caregivers in both sites reported statistically

significant improvements in their children's Strengths and Difficulties Questionnaire scores and improved parenting and family adjustment scores compared with baseline. The intervention in Afghanistan was delivered by a mix of mental health professionals and non-specialists, and the intervention in Serbia by non-specialists. Whilst the sample sizes were small and the follow up short-term, the interventions were deemed to have shown promise for brief, low cost, non-specialist mental health support for children and caregivers affected by conflict.

Four papers evaluated the impact of an intervention package to reduce girls' experience of violence and increase caregiver and social support networks to protect girls from violence. The intervention was implemented in the DRC, Ethiopia, and Pakistan [55,56,58,77], and included safe spaces, life skills training, mentorship, and a complementary caregiver programme. Outcomes in the three countries were evaluated by a mixed-methods study (Pakistan) and two randomised controlled trials (Ethiopia and DRC). In Pakistan, the intervention was found to have improved self-esteem, feelings of hope, and social support networks. At endline, participants were also more supportive of girls having the same opportunities as boys, and of girls working outside of the home after marriage [77]. In Ethiopia, participants demonstrated significantly improved social connectedness and school participation; however, there were no reported changes in girls' experience of sexual, physical or emotional violence [56]. Similarly, in the DRC, the intervention did not demonstrate an effect on reported experience of sexual, physical or emotional violence, neglect, transactional sex, or child marriage; however, significant improvements were observed in school participation, positive attitudes of caregivers towards gender equity, and increase caregiver warmth and affection [58]. Both the intervention group and the waitlist controls in DRC reported significant reductions in experience of physical and sexual violence at endline. In addition, increased gender-equity attitudes of caregivers was positively associated with school participation [55]. This series of studies highlights some nuances in the drivers of violence against children, the role of caregivers, and the role of context on children and young peoples' risk profiles.

Whilst the majority of interventions targeted the general population of children experiencing conflict, seven studies tested interventions to treat children who had screened positive for post-traumatic stress symptoms or psychiatric distress symptom [37,61,63,71,74–76]. All seven studies showed significant improvements in psychological symptoms after completion of treatment. The beneficial effects of the therapies were still evident at endline in the six studies that followed up after three months [61,63,74–76] and five months [37]. Three of the seven treatment interventions were delivered by mental health professionals or psychosocial support workers with training and experience in the provision of mental health and psychosocial support [37,75,76].

### Somatic paediatric and child health interventions

Nineteen (37%) of the 51 studies included a focus on one or more aspects of somatic child health. Nearly half of these were immunisation activities, including periodic intensification of routine immunisation (PIRI) and disease surveillance [52]; an integrated immunisation-nutrition intervention [53]; a mobile health intervention [41]; three vaccination campaigns [44,48,62]; a participatory learning and action intervention to increase vaccination uptake [54]; and a grassroots intervention to restore routine childhood immunisation during a humanitarian aid embargo [82]. All of the immunisation activities were found to have improved immunisation coverage. There were no studies on interventions for childhood noncommunicable diseases (NCDs).

The integrated nutrition-immunisation intervention observed lower vaccination dropout and fewer missed vaccinations in children who received immunisations in outpatient therapeutic feeding centres compared with children receiving vaccinations in health centres [53]. A mobile health outreach intervention delivered by local non-governmental organisations (NGO) was successful in vaccinating 51,168 previously unvaccinated children in hard to reach districts in Somalia over a period of 10 months, despite periodic service disruption due to insecurity [41]. The study did not report on vaccination of under-immunised children, or immunisation coverage. Whilst the three vaccination campaigns, the participatory learning and action intervention, the grassroots immunisation activity, and the PIRI intervention led to improvement in vaccination

coverage, only one campaign achieved sufficient coverage for herd immunity. The campaign was a government-led, multi-agency response to a vaccine-derived polio (cVDPV2) outbreak in Sudan in 2020 [48].

Adolescent sexual and reproductive health (SRH) interventions (N = 3/51) were focused on access to care, quality of care, child protection and MHPSS. A study in the DRC described contextual and age-specific patterns in experience of sexual violence and access to care at health facilities delivering sexual violence services in two sites, one post-conflict and one experiencing active conflict [46]. In the post-conflict site, patients accessing care were predominantly adolescents and young people (median age 15 years). The study found that sexual violence was more often perpetrated by civilians known to the survivor, patients were more likely to present for care after 72 hours (outside of the timeframe for optimally effective prophylaxis against HIV and sexually transmitted infections), and 38% had associated trauma. The main reason for delayed presentation was fear. In the site with active conflict, facilities received predominantly adult patients and the study outcomes were not disaggregated by age. A study of access to SRH services in Cameroon identified barriers in access to services for young people aged 10–24 years [36]. Young people described barriers to care, including a lack of awareness of available services or how and when to access them, shame, stigma, and fear, insecurity, while health workers cited a lack of expertise in adolescent SRH combined with services that were not designed to serve young people. The Congolese and Cameroonian studies did not report outcomes from the interventions. Adolescent participants in a study of a mobile GBV intervention in Lebanon reported that participation in psychosocial support activities improved their social connectedness, improved their emotional support and regulation, and increased their sense of safety and confidence in public spaces [67].

One study measured alterations in physiological stress responses in relation to an eight-week psychosocial intervention for refugee and host population adolescents [70]. The intervention was implemented in Adolescent Friendly Spaces in Jordan with age-matched and where possible, gender-matched group activities including fitness, arts and crafts, vocational skills, and technical skills. Intervention participants showed significant reduction in hair cortisol production (a marker of neuroendocrine stress response) across the 11-month study period (p = 0.005). The intervention demonstrated no effect on c-reactive protein levels or Epstein Barr Virus antibody titres.

Telemedicine interventions to support colleagues with limited paediatric expertise providing health care to children in settings characterised by insecurity and limited access for humanitarian actors were described in two studies. The interventions used different technologies to connect with colleagues in Syria (Skype and Facebook) [65] and Somalia (Audio-soft) [60]. The study in Somalia evaluated the changes to treatment and associations with patient outcomes compared with the prior year. The intervention, which included daily scheduled support for paediatric services, found that telemedicine support was associated with significant changes to clinical management in 64% of cases (e.g., changes to treatment regimen or medication dosage), and that 25% of cases referred to telemedicine included a life-threatening condition that was initially missed but later diagnosed with telemedicine support. Over time, management of potentially complicated cases was observed to have improved, evidenced by a linear reduction in telemedicine-directed changes to treatment for meningitis and convulsions (p = 0.001), lower respiratory tract infection (p = 0.02) and complicated malnutrition (p = 0.002). The intervention reduced adverse outcomes compared to the previous year (OR 0.70, 95% CI: 0.57–0.88, p = 0.001). The study reported that 45 patients needed to be treated through telemedicine to prevent one adverse outcome (defined as death or loss to follow up). The other telemedicine intervention documented successful provision of paediatric intensive care and nephrology support for a range of clinical decisions, including real-time support for resuscitation using telemedicine [65]. However, the descriptive design did not enable attribution of outcomes to the intervention.

### Child development and disability

One intervention specifically targeted child development. Four studies incorporated one or more measures of intervention impact on child development. A 10-day play-based intervention for Rohingya refugee children aged 5–12 years aimed to support the development of prosocial behaviour and executive functioning [78]. The intervention found improved prosocial

behaviours, with the patterns of change differing according to age and place of birth (Myanmar vs refugee camp). The other four studies described child friendly spaces (CFS) in multiple settings [50,66,84], and a microfinance programme with adolescents and caregivers in the DRC [40]. The evaluation of the microfinance programme had three study arms: adult caregivers only, integrated adolescent-caregiver, and adolescent only. For all three groups, children showed improved school attendance (p = 0.015) and prosocial behaviour scores (p = 0.032), with the greatest effects observed in the integrated parent-adolescent arm and the adolescent-only arm. The CFS interventions showed variable results. A meta-analysis of observational studies in five countries showed improved developmental assets in CFS attenders in Uganda (d = 0.37, 95% CI 0.15–0.59) and Iraq (d = 0.86, 95% CI 0.18–1.54), which disappeared on pooled analysis of data from the five sites [84]. One of the five sites was not a conflict- or post-conflict setting and accounted for over one-third of the pooled sample (N = 807, 35%). A study of a CFS intervention in Jordan (for children 6–17 years old) observed a decline in CFS attenders' developmental assets at endline and at 12-month follow up [66]. A CFS intervention in South Sudan (for adolescents 9–14 years old) found that both CFS attenders and waitlist controls had improved developmental assets at 12-month follow up, with CFS attenders showing significantly greater improvement compared with controls (p = 0.013) [50].

Of the 51 included studies, only two addressed an aspect of childhood disability. One intervention measured a mental health-related disability outcome in Syrian refugee children (aged 8–17 years old) receiving a telephone-based mental health intervention compared with treatment as usual with case management and psychotherapy delivered by an NGO [71]. Both groups had reduced disability scores over time, with no difference between the intervention group and the treatment as usual group. The other study was a participatory educational support group programme for the caregivers of children (aged 2–10 years) living with disability in Afghanistan [79]. The intervention consisted of nine sessions aimed at improving caregiver, child, and family well-being. Participants reported significant improvement in quality of life and family functioning at endline across all domains of the Paediatric Quality of Life, Family Impact Module (p < 0.0001). Caregivers reported changes that they attributed to the intervention, including moving from feeling frustration with their child to a feeling value and love for them. Parenting practices shifted correspondingly, with reports of increased patience, kindness and nurturing. Caregivers reported improved emotional self-regulation and described increased inclusive behaviours, including how they called the child ("mighty" vs "disabled"). Some caregivers reported that their children responded with increased participation in family and social life, improved mood, reduced social isolation, and improved mobility, communication, and self-care abilities after participating in the programme.

## Assessment of harm due to interventions

Only 10 of the 51 studies mentioned any assessment of whether the intervention caused harm to participants. In other words, 80% of published studies did not report whether any attempt was made to identify harm to children or families resulting from or related to their participation in the intervention. Only two studies reported adverse events. One was a vaccination campaign, which identified two cases of fever after vaccination; the patients were evaluated, managed symptomatically, and followed up [62]. The other study - of a PFA intervention - reported that some PFA providers had pressured participants to speak about difficult experiences and then failed to address the emotions and thoughts the participant had shared [72]. None of the remaining nine studies that assessed harm to participants reported adverse outcomes due to the interventions.

## Discussion

Our review identified 51 intervention studies, including several thoughtful studies that sought to address nuanced and upstream determinants of child health such as the role of caregivers and family, child and adolescent development, children's agency, and economic assets. Whilst the literature on child public health in conflict settings remains sparse, there is a notable increase in both the number of intervention studies published in recent years as well as the use of more rigorous

methods to evaluate interventions in conflict settings, particularly for MHPSS interventions. This is an improvement compared to the findings of a large, general systematic review of the effects of armed conflict on child health and development from 1945-2017, which identified mostly cross-sectional or descriptive studies [3].

Our review identified a range of promising MHPSS, parenting and child protection interventions, two-thirds of which were delivered by non-specialists. While nearly half of somatic paediatric interventions were immunisation activities, reports also described interventions for adolescent sexual and reproductive health, toxic stress, and telemedicine interventions to improve access to and quality of paediatric health services. The geographical distribution of interventions is similar to previous studies, with notable gaps in evidence for the Americas, North Africa, West Africa, South Asia, and Southeast Asia and the Pacific [3,33]. The geographical pattern of publications may reflect political framing of conflict, especially in Latin America [88]. Research focusing on the health impacts of armed conflict is relatively new, and prioritisation of funding for research in certain contexts is likely to influence the distribution of studies [3].

**First, do no harm**

It is concerning that only 10 of the 51 included studies mentioned any assessment of whether the intervention caused harm to participants. Among the 20 interventions that included a child protection component, only two studies mentioned assessment of harm to children from the intervention. These findings suggest that accountability to children in humanitarian response is a low priority; findings which are mirrored by the relative lack of mechanisms to enable accountability to children in the larger humanitarian health architecture [89]. The concern for harm to children caused by humanitarian interventions that were intended to support them is well founded. The risk is long recognised, and standards and guidance have been developed to prevent harm by humanitarian actors, with heavy focus on preventing exploitation and abuse [18,20]. There may also be unintended harms related to limitations in paediatric or child public health skills and knowledge, limited child-specific resources, and/or lack of adequate support and supervision. An example of the latter was seen in a study included in this review that identified instances where PFA providers encouraged people to speak about their traumas and then failed to support them afterward. A range of challenges relating to quality and safety of services for children have been described elsewhere [90].

Concerns about poor quality of care and barriers in access to health services for children in crisis contexts has led to a call for an intentional, informed and collaborative effort to improve child health services, beginning with a recognition of Humanitarian Paediatrics as a field of specialty within public health [90]. Two of the studies included in this review focused on telemedicine interventions, which aimed to do exactly that: improve the quality of paediatric care in settings with limited child health expertise, limited resources and lack of access due to insecurity. Whilst it does not address the underlying issues of resource distribution, telemedicine can be seen as a kind of palliative measure to improve existing services. Telemedicine carries a particular set of risks to patients that relate to patient confidentiality, the use of digital technology to collect and share sensitive personal information, and the potential to reinforce colonial power and resource differentials [91,92]. It is noteworthy that neither of the reports of telemedicine interventions mention assessment of risk, mitigation of risk, or occurrence of harms to children who received telemedicine support. One of the studies described using Skype and Facebook Messenger for communication with colleagues; the study took place before the General Data Protection Regulation (GDPR) came into effect, however it is important to note that this practice violates the GDPR [93]. Even carefully maintained and secured databases are vulnerable, as demonstrated by the 2022 cyber-attack on an International Committee of the Red Cross (ICRC) protection database [94].

**Improved evidence and persisting evidence gaps**

Over half of included studies and 95% of trials focused on MHPSS, child protection and/or parenting interventions, suggesting a growing demand for evidence to inform interventions in these sectors. A range of MHPSS, child protection, and

parenting interventions were successfully delivered to conflict-affected populations, some of which were living in inse-cure environments due to conflict. Non-specialists (e.g., doctors, clinical officers, nurses, midwives, or community health volunteers) delivered a large proportion of these interventions with supervision and support from mental health specialists, and most of these interventions showed promising results. These findings lend support to the current approach of larger foundational non-specialist interventions for basic needs, self-efficacy, and strengthening of community support, with respectively smaller scale non-specialised and specialised services [95]. The predominance of non-specialised MHPSS interventions is also consistent with a recent review of humanitarian health interventions that was not child-specific [96], however our review findings differed with respect to intersectoral interventions. We found that interventions for children and adolescents frequently took an intersectoral approach, combining MHPSS with elements of child protection and/or parenting support. However, this may simply reflect the humanitarian architecture, as child protection actors often take a leading role in MHPSS interventions [97].

Our review findings demonstrate that there is a long way to go to meaningfully incorporate the full breadth of child public health into humanitarian responses for children. Although studies of interventions focused solely on nutrition were excluded due to existing reviews on the topic, we included five studies reporting nutrition outcomes that were measured alongside other measures of child public health. The integration of nutrition with other study topics indicates a recognition of the importance of nutrition on child health. Additionally, the large proportion of interventions focused on parenting indicates a recognition of the role of caregivers and other adults in influencing child health and protection outcomes. Whilst this is encouraging, our review identified a lack of studies on childhood NCDs, child development, and disability interventions. The lack of studies on childhood NCDs is consistent with a recent systematic review on NCD interventions for women and children [98], and mirrors the lack of routine data collection on childhood NCDs by humanitarian actors in conflict settings [27]. Further, Munyuzangabo et al identified a lack of evidence for newborn health interventions [33]. Together, these findings indicate a lack of paediatric and child public health expertise among humanitarian actors.

Our review also identified a series of therapeutic MHPSS interventions for children and adolescents with recognised psychological distress or mental health disorders, which showed promising outcomes in the short-term. These therapeutic intervention studies indicate a growing prioritisation of mental health services during humanitarian response; this is an encouraging development, in light of the potentially life-changing and life-long harmful psychological impacts of conflict on children [6]. The frequent overlap of MHPSS, child protection, and parenting also indicates an increased focus on the social determinants of child health. With the existing funding and access challenges in conflict settings, it is unsurprising that most interventions were short-term or nested within interventions such as safe spaces. There is a need to identify solutions for continuity of therapeutic MHPSS services that are safe, effective, feasible, and sustainable in the long-term. Whilst the success of non-specialist interventions lends support for the use of these interventions in conflict settings, non-specialist services are unlikely to be sufficient for all patients, and risks of harm were identified that should be pre-vented or mitigated.

Several studies reported on interventions in baby-, child-, and adolescent-friendly spaces and safe spaces. The vari-able findings of these studies is consistent with other studies on the effectiveness of CFS [99]. Safe and friendly spaces are a mainstay of humanitarian response for children and provide an opportunity to support children's wellbeing, identify health and protection needs, and link them with relevant services. Studies and reviews on CFS and safe spaces under-score the fundamental role of context in influencing outcomes, and the importance of safety and quality of services to effectively support and protect children [99]. The lack of evidence on long-term benefits does not negate the value of friendly and safe spaces for children and young people. From a child public health approach, even short-term benefits to wellbeing and functional literacy may serve to mitigate the toxic stress impacts of severe adversity on children's bodies and brains. The study by Panter-Brick et al demonstrates the potential to mitigate toxic stress by an MHPSS intervention in a Child Friendly Space [70].

## Development and disability

Five of the included studies explored child and adolescent development using a mental health approach. No somatic studies explored child development either as an outcome or predictor of child physical health. Child and adolescent development is a major gap in humanitarian health response [100], and the lack of somatic paediatric or child health framing reflects the previously-described lack of both paediatric and child public health expertise amongst actors working in humanitarian settings [90]. This gap in expertise was also demonstrated in a recent scoping review of child public health indicators in fragile, conflict-affected and vulnerable settings [27]. The review did not identify a single child development indicator that is routinely measured by five large operational NGOs; recommended by technical agencies and partnerships; or required by donor agencies [27]. Our initial search identified one study describing three case studies on the implementation of Nurturing Care in conflict-affected children [101], however the study was excluded because it did not report data on children's or caregivers' outcomes. Nurturing Care is an approach to mitigate toxic stress and promote healthy child development through focus on good health, adequate nutrition, safety and security, responsive caregiving and providing opportunities for early learning [102]. A study of a CFS intervention demonstrated that harmful physiological responses to conflict-related stressors in adolescents can be treated effectively [70]. Toxic stress is well-described in the paediatric literature and has important implications for children and young people's physiological, neurological, and social development in the immediate-, short-, medium- and long-term [103]. Whilst the evidence on child and adolescent development in conflict settings remains limited, the studies included in this review make some progress towards documenting this foundational aspect of child health and wellbeing.

Childhood disability remains largely out of focus, with only one study of an intervention to support children with disability [79], and another study which included a measurement of disability related to mental health [71]. There were no interventions on injury care or rehabilitation. The limited focus on childhood disability is also seen in data collection by humanitarian public health actors [27]. The lack of data on disabilities amongst conflict-affected children is deeply concerning. Disabling injuries and illnesses in conflict-affected children are well documented [3]. These impacts on children are not historical. From January – November 2024, an estimated 21,000 children in Gaza sustained war-related injuries, and one-fourth (N = 5,230) of these children require rehabilitation [104]. This equates to an average of 15 children in Gaza sustaining potentially disabling injuries every day – the Protection Cluster in the Occupied Palestinian Territory states that these numbers are thought to be underestimates [104]. Responding safely and effectively to potentially disabling injuries in conflict-affected children requires technical expertise in paediatric trauma, medical equipment that is suitable for children, and understanding of the context-specific determinants of health that may affect the child's healing and prognosis during care and after discharge [90]. There is an urgent need to improve our understanding of childhood disability in conflict settings and of interventions to support affected children in the short- and long-term. Similar to interventions on child and adolescent development, this will require reliable data. While the availability and quality of child public health data are limited, routine situation and outcome data on child public health are collected in most settings by humanitarian actors, demonstrating that it is feasible to collect routine data in most contexts despite of contextual and funding challenges [27].

The intervention by Evans et al [79] included in this review is an excellent example of using a child public health approach to protect and promote the health of children with disabilities. The intervention was designed in recognition that child health, development and safety are influenced by social dynamics within the home and in the community, perceived value of the child, caregiving practices, and caregiver mental health. Participants reported extraordinary improvements in the abilities of their children after the intervention. For example one child that was previously immobile began to move independently, and another child who was nonverbal began to speak and also play with other children. Such profound improvements in children's health after positive changes in the social environment and with loving caregiving are a familiar observation for paediatricians and are well-described in the paediatric literature, most notably among children adopted from orphanages [105]. There is a great, yet untapped potential to meaningfully support child health and

development for all children in conflict-affected settings, including the most severely impacted children, through the use of child public health.

## Limitations

The findings of this review are likely to be limited by reporting bias, including reduced reporting of negative and/or nonsignificant outcomes [106]. Further, many of the studies included in our review measured outcomes based on participant and/or caregiver self-report and are thus prone to response bias. In humanitarian settings, affected populations may feel pressured to participate in studies in order to receive aid, presenting both ethical challenges to research as well as a risk of both selection bias and response bias that may affect study findings. A number of studies included in the review were observational, and several did not include a baseline comparison, limiting the conclusions that can be drawn. Reporting of gender-disaggregated outcomes was variable and thus limited our ability to report on gender-specific outcomes. Finally, the geographical distribution of studies was characterised by a notable lack of studies both within and across regions, especially from the Americas, and in particular Central and South America, as well as West Africa, North Africa, South Asia, Southeast Asia and the Pacific. The geographical gaps limit the visibility of conflict-affected children in the missing areas, and the evidence for interventions to support children may not be generalisable to these contexts. Nevertheless, our review compiles the available recent evidence for child public health interventions in conflict-affected children and may serve as a resource for actors wishing to design and deliver clinical or public health interventions for children.

The review included studies published in the peer-reviewed literature and grey literature. This design likely only captures a fraction of the evidence on the effectiveness and/or impact of interventions that were delivered for children affected by armed conflict during the study period, as many intervention evaluations by implementing and donor organisations are not shared publicly. Whilst our review had no language restrictions, we did not search region-specific databases, which may have limited the number of retrieved records and influenced the geographical distribution of identified studies. The search update did not include searches of the Global Health database or the grey literature, which may also have limited the number of retrieved studies. Lastly, the quality of studies and the certainty of evidence were not assessed, thus limiting the conclusions that can be drawn about the design and impact of the interventions. However, refraining from assessing quality and the certainty of evidence served to reduce bias towards better-resourced studies in more stable settings and improve the external validity of our review.

## Conclusion

The evidence for child public health interventions to support conflict-affected populations remains limited, but the numbers of studies and rigour of study design is improving. MHPSS, child protection, and parenting interventions show promise, and the increasing use of intersectoral approaches for these three thematic areas is encouraging. There are few studies of interventions to support somatic child health, particularly child and adolescent development, NCDs of childhood, and childhood disabilities. Geographical disparities in evidence persist within regions and globally. Few studies mention whether any assessment was made of harm to participants either caused by or related to the interventions. Given the challenges in delivering humanitarian responses for children, and the well-recognised risk of harm due to humanitarian aid, the lack of focus on safety is deeply concerning. Whilst the majority of intervention studies show promising results, the lack of data about safety suggests that conclusions about effectiveness should be treated with caution. There is an urgent need for evidence on the safety of interventions, medium- and long-term impacts, and interventions to support child and adolescent development, children with disability, and childhood NCDs. Research should be prioritised in geographical areas where data are lacking, and reporting should be age- and gender-disaggregated. The publication of nonsignificant and/or negative outcomes should be encouraged, as these findings provide valuable insights for humanitarian actors delivering interventions for children and families. Finally,

Humanitarian Paediatrics should be established as a profession within public health to support safe and effective child public health interventions in conflict-affected populations.

## Supporting information

**S1 Text. PRISMA checklist.**
(DOCX)

**S2 Text. Analytic codes.**
(DOCX)

**S1 Table. Template data extraction form.**
(XLSX)

## Author contributions

**Conceptualization:** Ayesha Kadir.

**Data curation:** Ayesha Kadir, Sneha Krishnan, Catherine McGowan, Daniel Martinez Garcia.

**Formal analysis:** Ayesha Kadir, Daniel Martinez Garcia.

**Investigation:** Ayesha Kadir.

**Methodology:** Ayesha Kadir, Sneha Krishnan, Catherine McGowan.

**Project administration:** Ayesha Kadir.

**Resources:** Catherine McGowan.

**Supervision:** Ayesha Kadir.

**Validation:** Ayesha Kadir, Daniel Martinez Garcia.

**Visualization:** Ayesha Kadir.

**Writing – original draft:** Ayesha Kadir.

**Writing – review & editing:** Sneha Krishnan, Catherine McGowan, Daniel Martinez Garcia.

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
