## [Decision Letter · Decision Letter 0]

6 Aug 2025

PGPH-D-25-00986

Child public health interventions for conflict-affected populations: A systematic review

Dear Dr. McGowan,

Thank you for submitting your manuscript to PLOS Global Public Health. After careful consideration, we feel that it has merit but does not fully meet PLOS Global Public Health’s publication criteria as it currently stands. Therefore, we invite you to submit a revised version of the manuscript that addresses the points raised during the review process.

We look forward to receiving your revised manuscript.

Kind regards,

Nancy Angeline Gnanaselvam

Academic Editor

Journal Requirements:

Additional Editor Comments (if provided):

1. Introduction needs to be strengthened by adding citations on SDGs, UNICEF, UNHCR, Sphere guidelines and other relevant reports

2. Key words can be increased and relevant to article

3. Health needs of this study population- mental and physical issues during humanitarian settings, vulnerabilities due to age such as child abuse, child sexual abuse etc needs elaboration in introduction

4. Authors also need elaborate on current scenario of humanitarian setting in the world - Gaza etc in introduction

5. Please use PRISMA checklist to structure the paper

6. Search words and MESH needs to be mentioned. The mentioned search words do not seem to capture the studies mentioned

7. Figure 1 - can be made into multiple tables - physical health needs, mental health needs, social interventions, preventive interventions etc so that the readers can understand clearly (Same classification as in Table 2 can be used)

8. Or the studies can be classified into cross sectional, analytical, experimental

9. A flow chart to depict studies included, excluded, etc will be helpful

10. Discussion needs to be strengthened from the health systems, applicable for current geopolitical scenario - intersectional coordination and how international global health actors can help in these interventions. Concusions and recommendations, limitations with respect to statistics and robust data analysis tools need to be strong and specific

Reviewers' comments:

Reviewer's Responses to Questions

**Comments to the Author**

1. Does this manuscript meet PLOS Global Public Health’s publication criteria ? Is the manuscript technically sound, and do the data support the conclusions? The manuscript must describe methodologically and ethically rigorous research with conclusions that are appropriately drawn based on the data presented.

Reviewer #1: Yes

2. Has the statistical analysis been performed appropriately and rigorously?

Reviewer #1: N/A

3. Have the authors made all data underlying the findings in their manuscript fully available (please refer to the Data Availability Statement at the start of the manuscript PDF file)?

Reviewer #1: Yes

4. Is the manuscript presented in an intelligible fashion and written in standard English?

Reviewer #1: Yes

5. Review Comments to the Author

Reviewer #1: This well-written review explores an important topic and its findings will be of great value to people working in humanitarian paediatrics. The discussion is comprehensive and highlights gaps and areas for action.

A few minor comments:

• The paragraph starting on line 206 includes a description on who is accessing the intervention and when (for study reference 41) and a description of service barriers (for study reference 21) rather than describing the outcomes associated with the interventions. While this is of interest and can help inform service development it is not in keeping with the findings presented elsewhere in the article, which focus on intervention outcomes.

• The relevance of including ‘inspired by an intervention in Syria where informational leaflets were distributed with bread’ is unclear in the sentence on line 143.

• Consider adding the study reference to the ‘other telemedicine study’ at the end of the sentence on line 247.

I am glad to recommend this paper for publication.

6. PLOS authors have the option to publish the peer review history of their article (what does this mean? ). If published, this will include your full peer review and any attached files.

**Do you want your identity to be public for this peer review?** For information about this choice, including consent withdrawal, please see our Privacy Policy .

Reviewer #1: No

---

## [Decision Letter · Decision Letter 1]

21 Nov 2025

Child public health interventions for conflict-affected populations: A systematic review

PGPH-D-25-00986R1

Dear Dr McGowan,

We are pleased to inform you that your manuscript 'Child public health interventions for conflict-affected populations: A systematic review' has been provisionally accepted for publication in PLOS Global Public Health.

Best regards,

Julia Robinson

Executive Editor

Reviewer Comments (if any, and for reference):

Reviewer's Responses to Questions

**Comments to the Author**

1. If the authors have adequately addressed your comments raised in a previous round of review and you feel that this manuscript is now acceptable for publication, you may indicate that here to bypass the “Comments to the Author” section, enter your conflict of interest statement in the “Confidential to Editor” section, and submit your "Accept" recommendation.

Reviewer #1: All comments have been addressed

2. Does this manuscript meet PLOS Global Public Health’s publication criteria ? Is the manuscript technically sound, and do the data support the conclusions? The manuscript must describe methodologically and ethically rigorous research with conclusions that are appropriately drawn based on the data presented.

Reviewer #1: Yes

3. Has the statistical analysis been performed appropriately and rigorously?

Reviewer #1: Yes

4. Have the authors made all data underlying the findings in their manuscript fully available (please refer to the Data Availability Statement at the start of the manuscript PDF file)?

Reviewer #1: Yes

5. Is the manuscript presented in an intelligible fashion and written in standard English?

Reviewer #1: Yes

6. Review Comments to the Author

Reviewer #1: Thank you for submitting the updated paper. I think the paper has been strengthened by the edits.

7. PLOS authors have the option to publish the peer review history of their article (what does this mean? ). If published, this will include your full peer review and any attached files.

**Do you want your identity to be public for this peer review?** For information about this choice, including consent withdrawal, please see our Privacy Policy .

Reviewer #1: No
